# FASTopoWM: Fast-Slow Lane Segment Topology Reasoning with Latent World Models

## Abstract

Lane segment topology reasoning provides comprehensive bird's-eye view (BEV) road scene understanding, which can serve as a key perception module in planning-oriented end-to-end autonomous driving systems. Current approaches prioritize graph modeling, endpoint alignment, and multi-attribute learning, yet they often neglect temporal modeling. This leads to inconsistent inter-frame detection within scene flows and motivates our focus on temporal propagation for lane segments. Recently, stream-based methods have shown promising outcomes by integrating temporal cues at both the query and BEV levels. However, it remains limited by over-reliance on historical queries, vulnerability to pose estimation failures, and insufficient temporal propagation. To overcome these limitations, we propose FASTopoWM, a novel fast-slow lane segment topology reasoning framework augmented with latent world models. To reduce the impact of pose estimation failures, this unified framework enables parallel supervision of both historical and newly initialized queries, facilitating mutual reinforcement between the fast and slow systems. Furthermore, we introduce latent query and BEV world models conditioned on the action latent to propagate the state representations from past observations to the current timestep. This design substantially improves the performance of temporal perception within the slow pipeline. Extensive experiments on the OpenLane-V2 benchmark demonstrate that FASTopoWM outperforms state-of-the-art methods in both lane segment detection and centerline perception. Our code will be released.

## 1 Introduction

Lane segment topology reasoning predicts lane segments (including centerlines and boundary lines) along with their topological relationships to construct a comprehensive road network (Wang et al., 2024a; Li et al., 2023b). This capability can be integrated into planning-oriented end-to-end autonomous driving systems, serving as the perception module to provide bird's-eye view (BEV) road scene understanding (Hu et al., 2023b; Jiang et al., 2023; Zhou et al., 2025).

Current methods primarily focus on constructing topology graphs (Lv et al., 2025), endpoint alignment (Fu et al., 2025a), and multi-attribute modeling (Li et al., 2023b). These methods enhance the topological associations between lane-to-lane and lane-to-traffic, as well as the long-range lane segment detection performance. However, they fail to leverage temporal information, leading to inconsistent detection across scene flows. Therefore, this paper focuses on achieving temporal perception for lane segments. Existing stream-based temporal propagation methods (Yuan et al., 2024) have proven effective in enhancing temporal consistency in perception. They reuse high-confidence historical

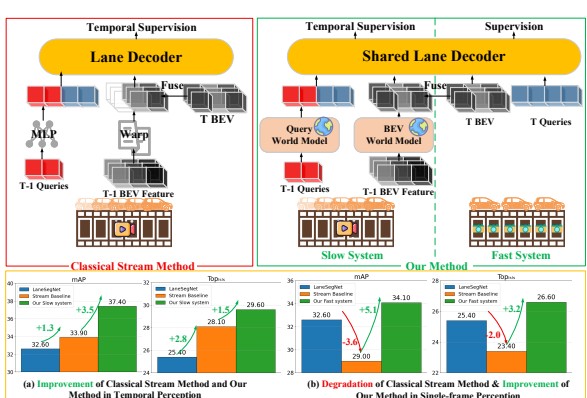

**Figure 1: Pipeline Comparison.** Existing stream-based methods suffer significant performance degradation when pose estimation is unavailable. Our approach addresses this issue by incorporating fast-slow pipelines and two latent world models.

queries and BEV features as anchors for current detection. However, as illustrated in Fig. 1, current stream-based frameworks suffer from three critical limitations: **(1) Over-reliance on historical queries.** The historical queries demonstrate higher confidence levels, they are more likely to approximate the GT positions compared to the newly initialized queries. As a result, during Hungarian matching supervision, historical queries are prioritized while newly initialized queries are often neglected. However, in the first frame of a scene, only the newly initialized queries are available for lane segment detection. If their performance is suboptimal, errors may propagate and accumulate across subsequent frames. **(2) Vulnerability to pose estimation failures.** To align historical queries and BEV features with the current frame, stream-based approaches rely on continuous pose estimation between adjacent frames. However, when vehicles enter tunnels or remote areas where GPS signals are unavailable, or when IMU error accumulation prevents pose refinement, the system degrades to single-frame detection mode. This degradation leads to significant performance deterioration and may ultimately cause complete system failure. **(3) Weak temporal propagation.** When historical BEV features are warped, the details of the features at the edges of the BEV tend to be lost. Moreover, simply applying pose transformation to historical queries through a basic MLP architecture fails to achieve optimal performance.

To address the aforementioned issues, we propose a novel **fa**st-**s**low lane segment **topo**logy reasoning framework with latent **w**orld **m**odels (**FASTopoWM**). Inspired by recent vision-language models (VLMs) (Zhang et al., 2025; Xiao et al., 2025), we decouple our network into dual pathways: a slow pipeline and a fast pipeline. The slow pipeline leverages temporal information to address challenging perception scenarios and improve detection performance, while the fast pipeline performs single-frame perception to ensure the basic functionality of the system. These two systems can perform inference in parallel or operate independently. A key innovation lies in the unified framework between the fast and slow systems that allowing parallel supervision of both historical and initialized queries. This eliminates the need for training separate models for temporal and single-frame detection and enables mutual reinforcement between the fast and slow pipelines. Specifically, the slow pipeline benefits from a better initialization for temporal propagation, while the fast pipeline implicitly leverages the temporal dynamics learned through shared weights. To improve temporal propagation, we propose latent query and BEV world models based on the principle that *"the present represents a continuation of the past."* Conditioned on the relative poses in adjacent frames, both latent world models propagate state representations from historical observations to the current timestep, significantly enhancing the robustness of temporal propagation in the slow pipeline.

**Contributions:** (1) We identify severe performance degradation issues in existing stream-based methods. To address this, we propose **FASTopoWM**, a novel fast-slow framework augmented with latent world models for robust lane segment topology reasoning. (2) We introduce a unified fast-slow system that enables parallel supervision of both historical and newly initialized queries. This design facilitates mutual reinforcement and allows inference switching based on pose estimation conditions, thereby enhancing system robustness. (3) We design two latent world models that effectively capture temporal dynamics and enable strong temporal propagation. (4) FASTopoWM is evaluated on the OpenLane-V2 dataset (Wang et al., 2024a), achieving state-of-the-art performance in lane topology reasoning.

## 2 RELATED WORK

### 2.1 HD MAP AND LANE TOPOLOGY REASONING

Current high-definition (HD) map learning frameworks employ end-to-end detection pipelines to generate vectorized representations of map elements. VectorMapNet (Liu et al., 2023) formulates map elements as polyline to eliminate heuristic post-processing. MapTR (Liao et al., 2022; 2023) develops permutation-equivalent modeling for lane point sets. Mask2map (Choi et al., 2024) incorporates mask-aware queries and BEV features to enhance semantic understanding. MapDR (Chang et al., 2025) decomposes road networks into geometric, connectivity, and regulatory layers. For temporal modeling, StreamMapNet (Yuan et al., 2024) implements stream-based propagation. SQD-MapNet (Wang et al., 2024b) applies query denoising for BEV boundary consistency. Interaction-Map (Wu et al., 2025) achieves comprehensive temporal fusion through local-to-global integration. In contrast to HD map learning, lane topology reasoning primarily focuses on the topological relationships among lanes. TopoNet (Li et al., 2023a) establishes dual lane-to-lane and lane-to-traffic

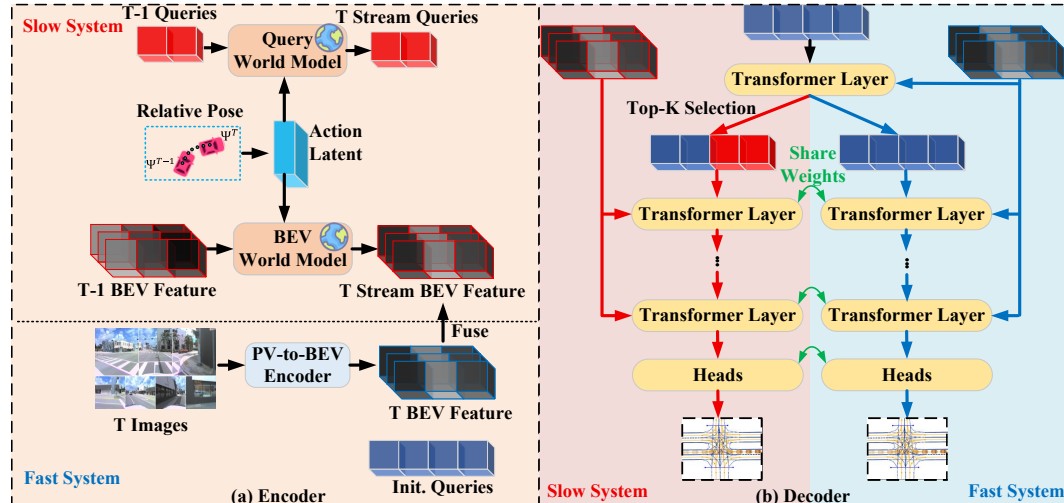

**Figure 2: Overall Framework. (a) Encoder.** Historical queries and BEV features are processed by world models, conditioned on action latent, to predict the stream queries and stream BEV features. Multi-view images are encoded into BEV features, which are then fused with the stream BEV features. **(b) Decoder.** The slow and fast systems share the same Transformer layers and prediction heads to enable parallel supervision of both stream and newly initialized queries. **T** represents the frame at timestep **T**.

graphs. TopoLogic (Fu et al., 2025a) and TopoPoint (Fu et al., 2025b) emphasize the critical role of endpoints in topological connectivity. TopoMLP (Wu et al., 2023) explicitly encodes coordinate information to improve topological reasoning. Topo2Mask (Kalfaoglu et al., 2024) introduces instance masks and mask attention mechanisms to update query representations. TopoFormer (Lv et al., 2025) proposes geometry-guided and counterfactual self-attention to enhance road topology understanding. TopoStreamer (Yang et al., 2025b) designs temporal propagation, denoising objectives, and positional encoding injection for multiple lane-segment attributes, enabling temporal perception of lane segments. Compared with TopoStreamer, our approach adopts a stronger dual world-model design for temporal propagation, using pose information as guidance to achieve more robust temporal perception. While TopoStreamer still relies on the traditional warping-based temporal propagation, it proposes streaming attribute constraints to ensure consistent propagation across multiple lane-segment attributes. It also introduces attribute-aware denoising objectives to improve the reliability of multi-pattern lane-segment associations. In addition, our paper identifies the significant performance degradation that occurs when temporal perception falls back to single-frame detection, and proposes a fast–slow system to enhance both temporal and single-frame detection performance. This design ensures system reliability in practical applications, and constitutes one of the key innovations that distinguishes our method from TopoStreamer.

## 2.2 WORLD MODEL IN AUTONOMOUS DRIVING

The world model serves as a bridge between understanding and generating future states. In autonomous driving, it can predict the future state of the ego vehicle conditioned on its actions (Bar et al., 2025). GAIA-1 (Hu et al., 2023a) leverages sequence learning to anticipate future events. DriveDreamer (Wang et al., 2024c) introduces a two-stage diffusion training pipeline to generate controllable driving scenes. Instead of synthesizing pixel-level content, OccWorld (Zheng et al., 2024) forecasts 3D occupancy to simulate future scenarios. To reduce computational overhead and modeling complexity, recent approaches have shifted toward predicting compact latent representations using world models. BEVWorld (Zhang et al., 2024) proposes a latent BEV sequence diffusion model to forecast future scenes conditioned on multi-modal inputs. WoTE (Li et al., 2025) predicts future BEV features to evaluate high-confidence trajectories. LAW (Li et al., 2024) employs a latent world model for self-supervised learning. Motivated by prior work, we design a dual latent world model framework that learns the transformation from past to present, enabling more effective temporal reasoning and enhancing current lane topology prediction.

## 2.3 FAST-SLOW SYSTEM

In autonomous driving scenarios, where real-time performance and reliability are essential, adding computational burden or auxiliary cues to boost prediction accuracy is often infeasible in practice. As a trade-off solution, fast-slow systems have recently been introduced. The fast system performs essential predictions with minimal overhead, while the slow system leverages additional computation and auxiliary cues to handle complex scenarios. Fast-slow systems enable flexible mode switching tailored to the varying demands of different scenarios. Chameleon (Zhang et al., 2025) utilizes a VLM with chain-of-thought (CoT) reasoning to conduct neuro-symbolic topology inference in the slow system. FAST (Xiao et al., 2025) dynamically switches between short and long reasoning paths based on task complexity. SlowFast-LLaVA (Xu et al., 2024) employs different sampling rates to capture motion cues and fuses slow and fast features to efficiently represent video information. Xu *et al.* (Xu et al., 2025) combine a "slow" LLM for command parsing with a "fast" RL agent for vehicle control. FASIONAD++ (Qian et al., 2025) incorporates a VLM as the slow system to provide feedback and evaluation for a fast end-to-end pipeline. Many existing approaches implement fast and slow systems as separate frameworks, switching between them based on the scenario. Our method, by contrast, integrates both pathways into a single model without requiring additional networks. These pathways are trained under parallel supervision. It is noted that the fast and slow systems reinforce each other, thereby enhancing overall robustness. Our architecture more closely resembles a unified fast-slow thinking model (Zhou et al., 2025; Xiao et al., 2025), where the inference pathway is switched by a trigger mechanism, such as scenario difficulty. For example, using slow thinking with reasoning capabilities in challenging scenarios, and fast thinking in simpler ones. In our case, the trigger is the status of the pose estimation; further discussion is provided in the supplementary material.

## 3 METHOD

### 3.1 PROBLEM FORMULATION

**Lane Segment Topology Reasoning.** Given surround-view images captured by vehicle-mounted cameras, our goal is to infer lane segments $\{\mathbf{L}^c, \mathbf{L}^l, \mathbf{L}^r\}$ and their topological connectivity $\mathbf{A}$ in the bird's-eye view (BEV). Each lane segment consists of a centerline $\mathbf{L}^c = (\mathbf{P}, Class)$, a left boundary $\mathbf{L}^l = (\mathbf{P}, Type)$, and a right boundary $\mathbf{L}^r = (\mathbf{P}, Type)$. Here, $\mathbf{P} = \{(x_i, y_i, z_i)\}|_{i=1}^M$ denotes a set of vectorized 3D points, where $M$ is the number of points. *Class* represents the semantic class of the lane segment (e.g., road line or pedestrian crossing), and *Type* indicates the boundary type, such as dashed, solid, or non-visible. The adjacency matrix $\mathbf{A}$ encodes the topological connectivity between lane segments. In the following formulation, $\mathbf{T}$ denotes the current frame, and $\mathbf{T\text{-}1}$ denotes the previous frame.

**Latent World Model.** Given the observation from camera $\mathcal{O}^{\mathbf{T}}$ at time T, the conventional world model forecasts the information about the next observation $\mathcal{O}^{\mathbf{T}+1}$. The details an be summarized as follows:

$$\mathbf{F}_{\mathbf{latent}}^{\mathbf{T}} = \mathcal{E}(\mathcal{O}^{\mathbf{T}}) \quad \mathbf{F}_{\mathbf{latent}}^{\mathbf{T}+1} = \mathcal{M}(\mathbf{F}_{\mathbf{latent}}^{\mathbf{T}}, \mathbf{Action}) \quad \mathcal{O}^{\mathbf{T}+1} = \mathcal{D}(\mathbf{F}_{\mathbf{latent}}^{\mathbf{T}+1}) \tag{1}$$

where $\mathcal{E}$ and $\mathcal{D}$ denote the encoder and decoder for the images, while the world model $\mathcal{M}$ maps the latent feature $\mathbf{F}_{\mathbf{latent}}^{\mathbf{T}}$ to $\mathbf{F}_{\mathbf{latent}}^{\mathbf{T}+1}$ based on the action. Compared to traditional driving world models, our approach has two key differences. First, we shift the prediction sequence of the world model backward along the timeline, enabling it to predict the current state based on historical data. Second, we employ a latent world model, eliminating the need to reconstruct observations from latent features. This allows the model to focus solely on temporal propagation in BEV space and query space, resulting in a more compact architecture and reduced learning complexity compared to image-based world models (Wang et al., 2024c; Hu et al., 2023a). Therefore, our latent world models follow the workflow of $\mathcal{O}^{\mathbf{T}-1} \rightarrow \mathbf{F}_{\mathbf{latent}}^{\mathbf{T}-1} \rightarrow \mathbf{F}_{\mathbf{latent}}^{\mathbf{T}} \rightarrow lane \quad segments$.

### 3.2 OVERVIEW

Fig. 2 illustrates the framework of **FASTopoWM**. The overall architecture can be briefly divided into encoder and decoder parts. Both the slow and fast systems are integrated into a unified network.

The PV-to-BEV encoder (Li et al., 2022; He et al., 2016; Lin et al., 2017) extracts BEV features $\mathbf{F}^{\mathbf{T}}_{\mathbf{bev}} \in \mathbb{R}^{H \times W \times C}$, where C, H, W represent the number of feature channels, height, and width, respectively. The initialized queries $\mathbf{Q}^{\mathbf{T}} \in \mathbb{R}^{N \times C}$ are derived from a learnable embedding space. $N$ is the number of queries. The relative ego poses between adjacent frames are encoded into an action latent $\Psi$. We employ memories for world models to store the historical queries. Conditioned on this action latent, the query world model and the BEV world model transform the historical queries $\mathbf{Q}^{\mathbf{T-1}}$ and BEV features $\mathbf{F}^{\mathbf{T-1}}_{\mathbf{bev}}$, respectively, to generate the stream queries $\tilde{\mathbf{Q}}^{\mathbf{T}}$ and stream BEV features $\tilde{\mathbf{F}}^{\mathbf{T}}_{\mathbf{bev}}$ for the current frame. The stream BEV features are then fused with the BEV features extracted from the current frame. The first transformer layer takes the initialized queries $\mathbf{Q}^{\mathbf{T}}$ and the BEV features $\mathbf{F}^{\mathbf{T}}_{\mathbf{bev}}$ from the current frame as input. The subsequent transformer layers share weights and receive parallel inputs from both the slow and fast systems. The slow system incorporates temporal information, whereas the fast system does not. During training, both systems are supervised jointly. At inference time, the prediction from the slow system is used as the final output when reliable pose information is available. Otherwise, when such information is missing or inaccurate, the prediction from the fast system is adopted. This design improves robustness while leveraging temporal cues to enhance performance when conditions permit.

### 3.3 TEMPORAL PROPAGATION VIA LATENT WORLD MODELS

Temporal propagation leverages detection results from previous frames as auxiliary information to assist prediction in the subsequent frame (Yuan et al., 2024). In contrast to existing latent world model methods (Li et al., 2024), our approach eliminates the need for future trajectory prediction. Instead, grounded in the principle that "the present represents a continuation of the past," we utilize known relative poses to infer the current state based on past observations.

**Input of World Models.** We flatten the relative pose, which includes both relative rotation and translation, to obtain the action latent $\Psi$. Then, the action-aware query and BEV latent features are obtained by:

$$
\begin{aligned}
\tilde{\mathbf{Q}}^{\mathbf{T-1}} &= \mathbf{MLP}([\mathbf{Q}^{\mathbf{T-1}}, \Psi]) \\
\tilde{\mathbf{F}}^{\mathbf{T-1}}_{\mathbf{bev}} &= \mathbf{MLP}([\mathbf{F}^{\mathbf{T-1}}_{\mathbf{bev}}, \Psi])
\end{aligned}
\tag{2}
$$

where $\Psi$ is duplicated and concatenated with the query and BEV feature along the channel dimension.

**Future Latent Prediction.** Latent world models leverage the historical action-aware latent from timestep **T-1** to predict the stream features at the next timestep **T**:

$$
\begin{aligned}
\tilde{\mathbf{Q}}^{\mathbf{T}} &= \mathbf{QueryWorldModel}(\tilde{\mathbf{Q}}^{\mathbf{T-1}}) \\
\tilde{\mathbf{F}}^{\mathbf{T}}_{\mathbf{bev}} &= \mathbf{BEVWorldModel}(\tilde{\mathbf{F}}^{\mathbf{T-1}}_{\mathbf{bev}})
\end{aligned}
\tag{3}
$$

where query world model is composed of Transformer blocks, containing self-attention modules and feed-forward modules. Similarly, the BEV world model comprises Transformer blocks that incorporate temporal self-attention modules (Li et al., 2022) and feed-forward modules.

Then, the stream BEV feature are fused with extracted BEV feature using gated recurrent unit (Chung et al., 2014) to enrich temporal cues.

**Future Latent Supervision.** Unlike previous methods that rely on warping and often lose information at the BEV boundary (Wang et al., 2024b), our BEV world model imagines the evolution of the next-frame BEV representation based on historical BEV features and relative poses. Since dense BEV annotations are difficult to obtain, we adopt a self-supervised learning strategy based on BEV features from temporally adjacent frames. We use mean squared error (MSE) loss to align the stream BEV features with the extracted BEV features of the current frame:

$$
\mathcal{L}_{bev} = \left\| \tilde{\mathbf{F}}^{\mathbf{T}}_{\mathbf{bev}} - \mathbf{F}^{\mathbf{T}}_{\mathbf{bev}} \right\|_2
\tag{4}
$$

For queries, we employ transformation loss (Yang et al., 2025b) to supervise consistency of coordinate, category, and semantic BEV mask:

$$\mathcal{L}_{coord} = \mathcal{L}_{L1}(\tilde{\mathbf{L}}_T, \mathbf{L}_T)$$
$$\mathcal{L}_{cls} = \mathcal{L}_{Focal}(\tilde{Class}_T, Class_T) + \mathcal{L}_{CE}(\tilde{Type}_T, Type_T)$$
$$\mathcal{L}_{mask} = \mathcal{L}_{CE}(\tilde{\mathbf{M}}_T, \mathbf{M}_T) + \mathcal{L}_{Dice}(\tilde{\mathbf{M}}_T, \mathbf{M}_T) \quad (5)$$
$$\mathcal{L}_{query} = \mathcal{L}_{coord} + \mathcal{L}_{cls} + \mathcal{L}_{mask}$$

where $\tilde{\mathbf{L}}_T$, $\tilde{Class}_T$, $\tilde{Type}_T$ and $\tilde{\mathbf{M}}_T$ are coordinates of lane segment, classes of centerline, boundary types and semantic BEV mask predicted from stream queries $\tilde{\mathbf{Q}}^{\mathbf{T}}$. $\mathbf{L}_T$, $Type_T$, $Class_T$ and $\mathbf{M}_T$ are GT annotations transformed from T-1 frame to T frame. For brevity, we omit the weights for each loss term. More details can be found in appendix. The overall future latent supervision can be expressed as:

$$\mathcal{L}_{latent} = \mathcal{L}_{bev} + \mathcal{L}_{query} \quad (6)$$

In this way, the BEV world model is trained using self-supervised learning on BEV representations from adjacent frames to capture temporal cues in the BEV space. Simultaneously, the query world model is trained with temporally continuous annotations, enabling it to transform historical observations into reference anchors (e.g., stream queries) for the current frame.

### 3.4 Unified Fast-Slow Decoder

Previous methods incorporate historical information into the decoder to improve performance. However, stream queries enriched with historical cues typically exhibit higher confidence than newly initialized queries. Consequently, during Hungarian assignment, GT annotations are more likely to be matched with stream queries. This bias leads to performance degradation when historical information is unavailable, as the model relies solely on the initialized queries. To overcome this limitation, we propose a unified fast-slow decoder that decouples the forward path into fast and slow branches, allowing parallel supervision of both stream and initialized queries.

As shown in Fig. 2 (b), the inputs of the first transformer layer are initialized queries and extracted BEV feature:

$$\mathbf{Q_1^T} = \mathbf{TransLayer_0}(\mathbf{Q_0^T}, \mathbf{F}_{bev}^T) \quad (7)$$

where $\mathbf{TransLayer_0}$ denotes the first transformer layer, consisting of a self-attention module, a lane-attention module (Li et al., 2023b), and a feed-forward network. According to the classification confidence of $\mathbf{Q_1^T}$, the lowest-ranked $N-K$ queries of $\mathbf{Q_1^T}$ are substituted with stream queries $\tilde{\mathbf{Q}}^{\mathbf{T}}$. The slow branch of the remaining transformer layer is then formulated as:

$$\tilde{\mathbf{Q}}_{\mathbf{i+1}}^{\mathbf{T}} = \mathbf{TransLayer_i}(\tilde{\mathbf{Q}}_{\mathbf{i}}^{\mathbf{T}}, \tilde{\mathbf{F}}_{\mathbf{bev}}^{\mathbf{T}}) \quad (8)$$

where $\mathbf{i}$ denotes the index of the Transformer layer. Notably, the input query to the second Transformer block, $\tilde{\mathbf{Q}}_{\mathbf{1}}^{\mathbf{T}}$, comprises $K$ instances of $\tilde{\mathbf{Q}}^{\mathbf{T}}$ and $N-K$ instances of $\mathbf{Q_1^T}$. For brevity, we reuse the notation $\tilde{\mathbf{F}}_{\mathbf{bev}}^{\mathbf{T}}$ to represent the fused BEV features obtained by combining the stream BEV features $\tilde{\mathbf{F}}_{bev}^{\mathbf{T}}$ with the extracted BEV features $\mathbf{F}_{bev}^{\mathbf{T}}$. Similarly, the fast branch of the second Transformer layer is formulated as:

$$\mathbf{Q}_{\mathbf{i+1}}^{\mathbf{T}} = \mathbf{TransLayer_i}(\mathbf{Q_i^T}, \mathbf{F}_{\mathbf{bev}}^{\mathbf{T}}) \quad (9)$$

Then, shared-weight heads are employed to generate predictions at each layer from $\mathbf{Q}_{\mathbf{i+1}}^{\mathbf{T}}$ and $\tilde{\mathbf{Q}}_{\mathbf{i+1}}^{\mathbf{T}}$, with Hungarian matching-based supervision applied in parallel.

In this manner, the historical queries and the newly initialized queries are supervised concurrently, preventing the model from becoming overly reliant on historical queries. The temporal propagation in the slow system benefits from well-trained initialized queries, particularly at the first frame. Meanwhile, the fast system's initialized queries implicitly gain from temporal dynamics through shared decoder parameters with the slow system. More details about prediction heads can be found in the appendix.

**Table 1:** Comparison with the state-of-the-arts on OpenLane-V2 subsetA on lane segment. All models adopt ResNet-50 as the backbone network and are trained for 24 epochs. $^{\dagger}$: Our enhanced model employ GeoDist strategy from TopoLogic (Fu et al., 2025a).

| Method | Venue | Temporal | mAP ↑ | $AP_{ls}$ ↑ | $AP_{ped}$ ↑ | $TOP_{lsls}$ ↑ | $Acc_b$ ↑ | FPS |
|---|---|---|---|---|---|---|---|---|
| MapTR (Liao et al., 2022) | ICLR23 | No | 27.0 | 25.9 | 28.1 | - | - | 14.5 |
| MapTRv2 (Liao et al., 2023) | IJCV24 | No | 28.5 | 26.6 | 30.4 | - | - | 13.6 |
| TopoNet (Li et al., 2023a) | Arxiv23 | No | 23.0 | 23.9 | 22.0 | - | - | 10.5 |
| LaneSegNet (Li et al., 2023b) | ICLR24 | No | 32.6 | 32.3 | 32.9 | 25.4 | 45.9 | 14.7 |
| TopoLogic (Fu et al., 2025a) | NIPS24 | No | 33.2 | 33.0 | 33.4 | 30.8 | - | - |
| Topo2Seq (Yang et al., 2025a) | AAAI25 | No | 33.6 | 33.7 | 33.5 | 26.9 | 48.1 | 14.7 |
| **FASTopoWM (ours)** | - | No | 34.1 | 33.9 | 34.4 | 26.6 | 48.2 | 14.0 |
| **FASTopoWM$^{\dagger}$ (ours)** | - | No | 34.2 | 34.0 | 34.4 | 28.4 | 48.2 | 14.0 |
| StreamMapNet (Yuan et al., 2024) | WACV24 | Yes | 20.3 | 22.1 | 18.6 | 13.2 | 33.2 | 14.1 |
| SQD-MapNet (Wang et al., 2024b) | ECCV24 | Yes | 26.0 | 27.1 | 24.9 | 16.6 | 39.4 | 14.1 |
| TopoStreamer (Yang et al., 2025b) | - | Yes | 36.6 | 35.0 | 38.1 | 28.5 | 50.0 | 13.6 |
| **FASTopoWM (ours)** | - | Yes | **37.4** | **36.4** | **38.4** | 29.6 | 51.2 | 11.4 |
| **FASTopoWM$^{\dagger}$ (ours)** | - | Yes | 37.2 | 36.2 | 38.1 | **31.6** | **51.3** | 11.2 |

## 4 TRAINING LOSS

The loss function for slow system and fast system are defined as:

$$\mathcal{L}_{slow} = \alpha_1 \mathcal{L}_{ls} + \alpha_2 \mathcal{L}_{latent} \tag{10}$$

$$\mathcal{L}_{fast} = \mathcal{L}_{ls} \tag{11}$$

where $\mathcal{L}_{ls}$ denotes the lane segment loss, which supervises the predicted lane segments using Hungarian matching (Li et al., 2023b). The details of $\mathcal{L}_{ls}$ can be found in the appendix.

The overall loss function in FASTopoWM is defined as follows:

$$\mathcal{L} = \mathcal{L}_{slow} + \mathcal{L}_{fast} \tag{12}$$

## 5 EXPERIMENTS

### 5.1 DATASETS AND METRICS

**Datasets.** We evaluate our method on the multi-view lane topology benchmark OpenLane-V2 (Wang et al., 2024a), which comprises two subsets. SubsetA is re-annotated from Argoverse2 (Wilson et al., 2023), while SubsetB is re-annotated

**Table 2:** Comparison with the state-of-the-arts on OpenLane-V2 subsetB on centerline perception. All models adopt ResNet-50 as the backbone network and are trained for 24 epochs. TopoFormer$^{\star}$ adopts a staged training strategy that utilizes a pretrained lane detector for topology reasoning training. While this leads to better detection performance, it offers only slight advantage in topology prediction.

| Method | Venue | Temporal | OLS ↑ | $DET_l$ ↑ | $TOP_{ll}$ ↑ |
|---|---|---|---|---|---|
| VectorMapNet (Liu et al., 2023) | ICML23 | No | - | 3.5 | - |
| STSU (Can et al., 2021) | ICCV21 | No | - | 8.2 | - |
| MapTR (Liao et al., 2022) | ICLR23 | No | - | 15.2 | - |
| TopoNet (Li et al., 2023a) | Arxiv23 | No | 25.1 | 24.3 | 6.7 |
| TopoMLP (Wu et al., 2023) | ICLR24 | No | 36.2 | 26.6 | 19.8 |
| LaneSegNet (Li et al., 2023b) | ICLR24 | No | 38.7 | 27.5 | 24.9 |
| TopoLogic (Fu et al., 2025a) | NIPS24 | No | 36.2 | 25.9 | 21.6 |
| TopoFormer$^{\star}$ (Lv et al., 2025) | CVPR25 | No | 41.5 | 34.8 | 23.2 |
| **FASTopoWM (ours)** | - | No | 41.8 | 31.6 | 27.1 |
| StreamMapNet (Yuan et al., 2024) | WACV24 | Yes | 26.7 | 18.9 | 11.9 |
| SQD-MapNet (Wang et al., 2024b) | ECCV24 | Yes | 29.1 | 21.9 | 13.3 |
| TopoStreamer (Yang et al., 2025b) | - | Yes | 42.6 | 30.9 | 29.4 |
| **FASTopoWM (ours)** | - | Yes | **46.3** | **35.1** | **33.0** |

from nuScenes (Caesar et al., 2020). Both subsets contain surround-view images collected from 1,000 scenes. SubsetA features seven camera views, whereas SubsetB includes only six. SubsetA provides annotations for both lane segments and their topology, while SubsetB contains annotations only for centerlines and topology. To generate boundary annotations for SubsetB, we apply a standardized lane width perpendicular to the centerline. For centerline perception evaluation on SubsetB, we approximate centerlines by averaging the coordinates of the left and right boundary lines. We re-train LaneSegNet, StreamMapNet, and SQD-MapNet with the same configuration to obtain their results on SubsetB.

**Metrics.** We conduct evaluations on two tasks: lane segment perception on SubsetA and centerline perception on SubsetB. To assess lane quality, we adopt Chamfer Distance and Fréchet Distance under fixed thresholds of {1.0, 2.0, 3.0} meters. For lane segments, $AP_{ls}$ and $AP_{ped}$ are employed to evaluate the detection performance of road lanes and pedestrian crossings, respectively. The mean average precision (mAP) is calculated as the average of $AP_{ls}$ and $AP_{ped}$. To evaluate topology

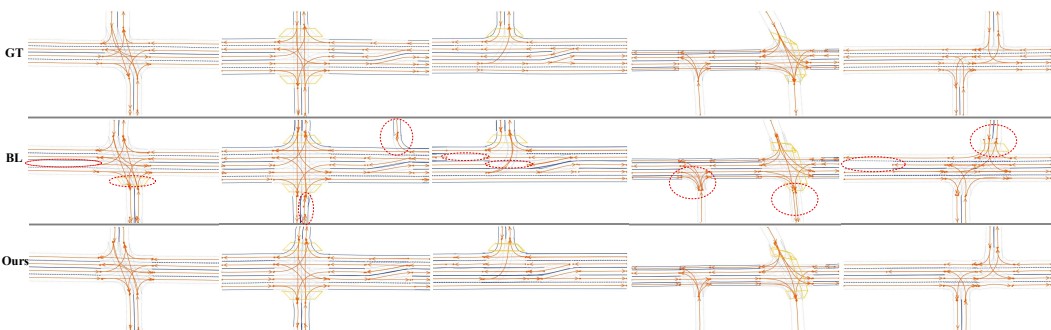

**Figure 3:** Qualitative results of baseline and our FASTopoWM. The baseline (BL) is LaneSegNet with stream-based temporal propagation. For better viewing, zoom in on the image.

reasoning, we report $\text{TOP}_{lsls}$. Lane boundary classification accuracy is measured by $\text{Acc}_b$ (Yang et al., 2025b). The evaluation protocol for centerline perception follows a similar approach as lane segments. Additionally, OLS (Wang et al., 2024a) is computed between $\text{DET}_l$ and $\text{TOP}_{ll}$.

## 5.2 IMPLEMENTATION DETAILS

The PV-to-BEV encoder is composed of a pre-trained ResNet-50 (He et al., 2016), an FPN (Lin et al., 2017), and BevFormer (Li et al., 2022). The BEV features has a resolution of $200\times100$, covering a perception area of $\pm50$m $\times \pm25$m. The decoder follows the Deformable DETR architecture, where the standard cross-attention module is substituted with lane attention (Li et al., 2023b). It consists of 6 transformer layers. A total of 200 queries are used,

**Table 3:** Ablation studies on different modules. The baseline is the non-temporal framework LaneSegNet. *Stream* refers to the baseline augmented with temporal propagation. *FS* denotes the fast-slow system. *QWM* and *BWM* represent the query world model and BEV world model, respectively. *Tem.* and *Sin.* indicate temporal and single-frame detection.

| Modules | | | | Tem. | | Sin. | |
|---|---|---|---|---|---|---|---|
| *Stream* | *FS* | *QWM* | *BWM* | mAP | $\text{TOP}_{lsls}$ | mAP | $\text{TOP}_{lsls}$ |
| | | | | - | - | 32.6 | 25.4 |
| ✓ | | | | 34.0 | 28.1 | 29.8 | 23.4 |
| ✓ | ✓ | | | 35.3 | 28.4 | 33.2 | 25.8 |
| ✓ | ✓ | ✓ | | 36.3 | 29.0 | 33.7 | 26.3 |
| ✓ | ✓ | | ✓ | 36.4 | 29.1 | 33.6 | 26.2 |
| ✓ | ✓ | ✓ | ✓ | **37.4** | **29.6** | **34.1** | **26.6** |

with 30% reserved for temporal propagation in the slow system. Both the query and BEV world models employ 2 transformer layers each. To optimize memory usage on the GPU, we introduce average pooling within the BEV world model to reduce the input BEV resolution from $200\times100$ to $100\times50$. Then, we use bilinear interpolation to restore the output resolution to $200\times100$. The centerline, left boundary, and right boundary are ordered sequence of 10 points. Training is performed over 24 epochs with a batch size of 8 on NVIDIA V100 GPUs. To stabilize the streaming process, the first 12 epochs are trained using single-frame inputs. More details about stream-based training can be found in the appendix. The learning rate is initialized at $2 \times 10^{-4}$ and follows a cosine annealing schedule throughout training. We use the AdamW optimizer (Kingma & Ba, 2015). The loss weights $\alpha_1$ and $\alpha_2$ are set to 1.0 and 0.3, respectively. During inference, the fast and slow systems can operate in parallel to generate predictions, or alternatively, inference can be performed using only one of the systems.

## 5.3 MAIN RESULTS

**Results on Lane Segment.** The results are displayed in Tab. 1. Without incorporating temporal information, our fast system achieves detection performance comparable to current state-of-the-art (SOTA) methods. Furthermore, our original and enhanced model establishes a new SOTA performance in mAP and topology accuracy.

**Results on Centerline Perception.** The results are presented in Tab. 2. Our fast system surpasses LaneSegNet by 3.1% in OLS. Our slow system outperforms TopoStreamer by 3.7% in OLS. By introducing world models to capture temporal information, our slow system achieves a 4.5% improvement in OLS over the fast system.

## 5.4 ABLATION STUDIES

The ablation studies are mainly conducted on SubsetA.

**Table 4:** The ablation studies of different configurations in the proposed FASTopoWM. The experiments are conducted on OpenLane-V2 subset A. We bold the best scores.

**(a)** Different training methods. The baseline is Lane-SegNet with stream-based temporal propagation.

| Method | Tem. | | Sin. | |
|---|---|---|---|---|
| | mAP | TOP$_{lsls}$ | mAP | TOP$_{lsls}$ |
| BaseLine | 34.0 | 28.1 | 29.8 | 23.4 |
| Random+WMs | 35.3 | 28.5 | 33.9 | 26.2 |
| FastSlow+WMs | **37.4** | **29.6** | **34.1** | **26.6** |

**(b)** Different action condition the world models.

| Action | Tem. | | | |
|---|---|---|---|---|
| | mAP | AP$_{ls}$ | AP$_{ped}$ | TOP$_{lsls}$ |
| None | 36.5 | 35.1 | 37.9 | 28.5 |
| Traj. | 31.2 | 30.8 | 31.5 | 24.5 |
| Pos. | **37.4** | **36.4** | **38.4** | **29.6** |

**(c)** Different architecture of world models.

| Arch. | Tem. | | | |
|---|---|---|---|---|
| | mAP | AP$_{ls}$ | AP$_{ped}$ | TOP$_{lsls}$ |
| Linear | 35.2 | 34.2 | 36.2 | 28.5 |
| MLPs | 36.2 | 35.9 | 36.2 | 28.9 |
| Transformer | **37.4** | **36.4** | **38.4** | **29.6** |

**(d)** Different number of layers in the world models.

| No. QWM | Tem. | | No. BWM | Tem. | |
|---|---|---|---|---|---|
| | mAP | TOP$_{lsls}$ | | mAP | TOP$_{lsls}$ |
| 2 | **37.4** | 29.6 | 2 | 37.4 | 29.6 |
| 4 | 37.1 | **29.9** | 4 | 37.7 | 29.9 |
| 6 | 37.3 | 29.5 | 6 | **37.8** | **30.2** |

**(e)** Different memory length of BEV world model.

| Length | Tem. | | | |
|---|---|---|---|---|
| | mAP | AP$_{ls}$ | AP$_{ped}$ | TOP$_{lsls}$ |
| 1 | 37.4 | 36.4 | 38.4 | 29.6 |
| 2 | 37.5 | 36.7 | 38.5 | 29.8 |
| 4 | **37.7** | 36.4 | **39.0** | 29.6 |
| 6 | 37.6 | **36.8** | 38.4 | 29.8 |
| 8 | 37.5 | 36.6 | 38.4 | **29.9** |

**(f)** Diferent number of queries for temporal propagation in query world model.

| TopK | Tem. | | | |
|---|---|---|---|---|
| | mAP | AP$_{ls}$ | AP$_{ped}$ | TOP$_{lsls}$ |
| 20 (10%) | 35.2 | 36.1 | 34.3 | 28.7 |
| 66 (30%) | **37.4** | **36.4** | **38.4** | **29.6** |
| 100 (50%) | 36.7 | 35.4 | 38.0 | 29.2 |
| 150 (75%) | 34.9 | 34.5 | 35.3 | 27.4 |

**(g)** Effect of latent supervision.

| Method | Tem. | | | |
|---|---|---|---|---|
| | mAP | AP$_{ls}$ | AP$_{ped}$ | TOP$_{lsls}$ |
| FASTopoWM (without latent.) | 36.6 | 35.8 | 37.5 | 29.0 |
| FASTopoWM (with latent.) | **37.4** | **36.4** | **38.4** | **29.6** |

**(h)** Different temporal propagation modules.

| Method | Tem. | | | |
|---|---|---|---|---|
| | mAP | AP$_{ls}$ | AP$_{ped}$ | TOP$_{lsls}$ |
| Baseline+Warp | 34.0 | 33.3 | 34.7 | 28.1 |
| Baseline+WMs | **36.5** | **35.6** | **37.4** | **28.9** |

**Ablation Study on Modules.** As shown in Tab. 3, integrating the baseline with temporal propagation improves performance by 1.4% mAP and 2.7% TOP$_{lsls}$. However, its performance of single-frame detection drops by 2.8% mAP and 2.0% TOP$_{lsls}$. This is because the temporal framework tends to over-rely on historical queries. When the lack of pose information makes temporal propagation unavailable, the model falls back to single-frame detection, where its localization capability is poor due to the reliance on the initialized queries as anchors. By integrating the fast-slow system with world models, we address the performance degradation issue of single-frame detection within temporal frameworks. This is because we decouple the supervision of initialized queries from that of historical queries, allowing the initialized queries to receive direct supervision from the ground truth. The proposed model achieves substantial improvements, surpassing the baseline by 3.4% and 1.5% mAP in temporal and single-frame detection, respectively.

**Ablation Study on Training Methods.** As shown in Tab. 4a, the baseline method shows a significant drop in single-frame detection performance. PrevPredMap (Peng et al., 2025) addresses this issue by randomly alternating between single-frame and temporal training modes. However, this alternating training reduces the effectiveness of temporal feature learning. In contrast, our fast-slow system enables parallel training of both modes, achieving improvements of 2.1% mAP and 1.1% TOP$_{lsls}$ in temporal detection.

**Ablation Study on World Models.** Tab. 4b examines the impact of different action conditions. Without action conditioning, the world model can still predict future states, but localization accuracy degrades. Inspired by end-to-end driving methods, we also condition on future trajectories, which provide future positions and could reduce reliance on pose estimation. However, this leads to clear performance drops, likely due to limited trajectory regression accuracy and the model's lack of agent-awareness and dynamics inputs (e.g., speed, steering). Conditioning on relative ego pose yields the best results. Tab. 4c compares network architectures: linear layers and MLPs fail to capture temporal dependencies, while stacked transformers achieve the best performance. As shown

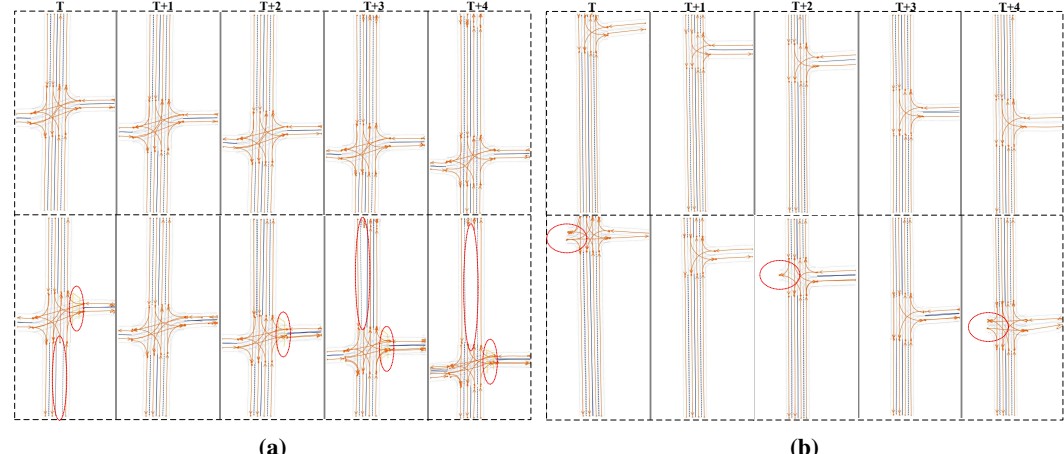

**(a)**  **(b)**

**Figure 4:** Visualization of topology predictions across consecutive 5 frames. The results of FASTopoWM are shown on the top, and the results of temporal baseline are shown on the bottom. The temporal baseline is LaneSegNet (Li et al., 2023b) with stream-based temporal propagation. For better viewing, zoom in on the images.

in Tab. 4d, we fix the number of transformer layers in one world model while varying it in the other. The model performance is relatively insensitive to the number of query world model layers. While deeper models offer slight gains, they reduce inference efficiency. We use two layers as a trade-off. In the Tab. 4e, we investigate the impact of different memory lengths in the BEV world model on performance. We transform BEV features extracted from longer-term historical frames to the current frame using the BEV world model, averaged them, and then fuse the result with the BEV feature of the current frame. The results show that increasing the memory length only yields marginal performance improvements. As shown in Tab. 4f, we explore the impact of the number of queries used for temporal propagation in the query world model. Experiments demonstrate that the configuration employed by FASTopoWM, which propagates the top 30% (i.e., 66) of queries, achieves the optimal results. Tab. 4g shows that employing the latent supervision from Eq. 6 yields an improvement of 0.8% mAP. Tab. 4h demonstrates that compared to traditional warping methods, our world models enable more robust temporal propagation, achieving a 2.5% improvement in mAP.

## 5.5 Qualitative Results

Fig. 3 provides a qualitative result comparison between baseline method and our FASTopoWM under different road structures. The baseline method produces more misaligned endpoints, which confuses topology reasoning. It also suffers from missed detections and hallucinated results. In contrast, our method yields good lane segment perception with accurate topology reasoning. Fig. 4 visualizes the comparison of temporal detection results across 5 frames. The baseline method fails to maintain temporal topological consistency, resulting in false detections, missed detections, and hallucinated topologies. Our method demonstrates robust temporally consistent topology reasoning results.

## 6 Conclusion

In this paper, we propose FASTopoWM, a novel fast-slow lane segment topology reasoning framework enhanced with latent world models. To overcome the limitations of existing stream-based methods, we integrate fast and slow systems into a unified architecture that enables parallel supervision of both historical and newly initialized queries, fostering mutual reinforcement. The slow pipeline exploits temporal information to enhance detection performance, while the fast pipeline conducts single-frame perception to ensure the system's basic functionality. To further strengthen temporal propagation, we introduce latent query and BEV world models conditioned on the action latent, allowing the system to propagate state representations from past observations to the current timestep. This design significantly boosts the performance of the slow pipeline. Extensive experiments on the OpenLane-V2 benchmark demonstrate that our model achieves SOTA performance and validate the effectiveness of our proposed components.

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

# A    APPENDIX

## A.1    POSE ESTIMATION

Poses provide dynamic information for autonomous vehicles and are typically obtained via GPS (Global Positioning System) and IMU (Inertial Measurement Unit). Specifically, GPS determines position through satellite-based ranging (absolute coordinates), while the IMU derives position and orientation by integrating acceleration and angular velocity (relative motion). The advantage of GPS is its absence of long-term cumulative errors, whereas the IMU offers high frequency, short-term stability, and continuity. The integration of these two methods for pose estimation has been widely adopted in many public datasets, like KITTI (Geiger et al., 2012), ApolloScape (Huang et al., 2018), Cityscapes (Cordts et al., 2016), H3D (Patil et al., 2019), etc. However, this approach also has limitations. For instance, in the KITTI dataset, the localization system is susceptible to GPS signal interruptions, particularly in urban canyons or tunnels where interference is common (Brossard et al., 2020). In such cases, the IMU can be used for compensation, but it suffers from error accumulation that increases exponentially over time (drift), resulting in poor long-term stability. In the nuScenes (Caesar et al., 2020), to mitigate this issue, carefully collected detailed high-definition LiDAR point cloud maps are employed. However, in practice, it is not feasible to create such point cloud maps for all scenarios.

*How to define missing or inaccurate pose?*

From a localization and sensor-fusion perspective, it refers to cases where the estimated vehicle position or orientation (pose) cannot be reliably determined due to signal loss, sensor drift, or inconsistency between different measurements. It can be characterized and measured differently for GPS and IMU systems.

**GPS.** Pose information is considered missing or inaccurate when the satellite-based positioning solution fails to meet required quality thresholds. In our practical experience, commonly observed issues include: **1. RTK (Real-Time Kinematic) status deteriorates from FIX → FLOAT → DGPS → Single.** This indicates a gradual degradation in positioning accuracy. **2. Poor geometric precision.** High HDOP (Horizontal Dilution of Precision)/PDOP (Position Dilution of Precision) values (e.g., HDOP $> 4$) reflect poor satellite geometry. **3. Weak or unstable satellite signals.** Low C/N$_o$ (carrier-to-noise ratio $< 30$ dB-Hz) or fewer than 6 visible satellites.

**IMU.** Pose is considered inaccurate when integration of accelerations and angular rates accumulates error beyond acceptable limits. Since IMU data are relative and drift over time, accuracy degrades when not corrected by GPS or other sensors. In our practical experience, commonly observed issues include: **1. Rapid drift of position or orientation.** A drift is identified if the position drift meets or exceeds 0.1 m per second without external correction, or if the yaw/roll bias accumulation meets or exceeds $0.1°$ per minute. **2. Dynamic inconsistency.** Pose-derived velocity or acceleration inconsistent with wheel odometry or CAN bus data. **3. Time synchronization error.**

*How to measure inaccurate pose?*

To evaluate the accuracy of the current pose estimated from IMU and GPS, one can perform consistency and residual checks across the three data sources—CAN bus (vehicle kinematics), IMU, and GPS. The core idea is to compare physical quantities derived from the estimated pose with independent measurements from the vehicle and sensors. **1. Consistency between IMU and CAN bus.** If the IMU is well calibrated, its measured yaw-rate and acceleration should match the CAN yaw-rate and acceleration within small tolerance. **2. GPS residual analysis.** When fusing GPS and IMU, we can validate pose accuracy by examining the residual between the fused result and the raw GPS measurement:

$$r = P_{GPS} - P_{fusion} \qquad (1)$$

where $P_{GPS}$ is the observed GPS position and $P_{fusion}$ is the predicted position obtained from IMU integration combined with the previous state estimate. We then determine whether the magnitude of r exceeds a predefined threshold.

*Trigger.* GPS plays a primary role in positioning because it also provides the reference needed to correct the IMU. Consider the following scenario: before entering a tunnel, GPS and IMU fusion

operates normally. Inside the tunnel, GPS becomes unavailable and the system relies solely on the IMU, causing the drift to grow over time. After exiting the tunnel, GPS becomes available again and corrects the accumulated IMU drift, restoring normal behavior. Therefore, we monitor both GPS signal quality and GPS residuals as triggers for detecting inaccurate pose.

In traditional temporal methods, when pose information becomes unreliable, the system must fall back to single-frame detection and stop temporal propagation. In this case, as validated in Table 3 of the main text, the performance of the single-frame detector drops by 2.8% in mAP and 2.0% in $\text{TOP}_{lsls}$ compared with a well-trained single-frame model. Alternatively, an autonomous driving system would need to carry both temporal and single-frame models, which introduces additional computational and memory overhead.

However, with our fast–slow architecture that shares weights and performs parallel supervision for both single-frame and temporal branches, this problem is resolved. Moreover, the single-frame detection performance is improved, achieving gains of approximately 1.5% in mAP and 1.2% in $\text{TOP}_{lsls}$. In addition, introducing the world model further boosts the temporal detection performance, improving the baseline temporal method by 3.4% in mAP and 1.5% in $\text{TOP}_{lsls}$.

## A.2 ACTION LATENT

In the ablation study, we try to use both relative pose and trajectory as conditions for the world models. Ultimately, conditioning on relative poses achieve the best performance.

**Related Pose.** The relative pose is defined as the transformation matrix that maps the rotation matrix R and translation vector t from the coordinate system of the previous frame to that of the current frame. It can be formulated as:

$$\mathbf{T}_{\text{prev}}^{\text{global}} = \begin{bmatrix} \mathbf{R}_{\text{prev}} & \mathbf{t}_{\text{prev}} \\ \mathbf{0} & 1 \end{bmatrix} \tag{2}$$

$$\mathbf{T}_{\text{global}}^{\text{related}} = \begin{bmatrix} \mathbf{R}_{\text{curr}}^{\top} & -\mathbf{R}_{\text{curr}}^{\top}\mathbf{t}_{\text{curr}} \\ \mathbf{0} & 1 \end{bmatrix} \tag{3}$$

$$\mathbf{T}_{\text{prev}}^{\text{curr}} = \mathbf{T}_{\text{global}}^{\text{related}} \cdot \mathbf{T}_{\text{prev}}^{\text{global}} \tag{4}$$

$$\Psi = \text{Flatten}(\mathbf{T}_{\text{prev}}^{\text{curr}}) \tag{5}$$

**Trajectory.** The trajectory represents the future positions of the ego vehicle. We extract positions over a 3-second horizon (equivalent to 6 frames) and transform them into the current BEV coordinate system to generate 6 waypoints $\mathbf{W}_t = \{\mathbf{w}_t^1, \mathbf{w}_t^2, \cdots, \mathbf{w}_t^6\}$ (Li et al., 2024). In the final five frames of a scene, some future positions may be unavailable; in such cases, interpolation is applied to complete the trajectory. The trajectory query is randomly initialized and refined via cross-attention with the BEV features and lane segment queries. A MLP is used to predict the trajectory from trajectory query. Then, the loss for trajectory prediction can be formulated as:

$$\mathcal{L}_{traj} = \frac{1}{M} \sum_{i=1}^{M} \left\| \mathbf{w}_t^i - \tilde{\mathbf{w}}_t^i \right\|_1 \tag{6}$$

We concatenate the predicted waypoints with the query and BEV features along the channel dimension, and use the world model to predict the state of the next frame.

## A.3 TRANSFORMATION LOSS

Transformation losses are applied to supervise both the stream query and the query world model, with the goal of minimizing projection errors across frame transitions. We employ MLPs to predict lane segment coordinate, lane segment class, boundary class and BEV mask from stream queries

$\tilde{\mathbf{Q}}^{\mathbf{T}}$:

$$\tilde{\mathbf{L}}_t^c = \text{MLP}_{reg}(\tilde{\mathbf{Q}}^{\mathbf{T}}) + \text{InSigmod}(\mathbf{R}_{\mathbf{C}}^{\mathbf{S}})$$

$$\tilde{\mathbf{L}}_t^c = \text{Denorm}(\text{sigmoid}(\tilde{\mathbf{L}}_t^c))$$

$$offset = \text{MLP}_{offset}(\tilde{\mathbf{Q}}^{\mathbf{T}})$$

$$\tilde{\mathbf{L}}_t^l = \tilde{L}_t^c + offset, \tilde{\mathbf{L}}_t^r = \tilde{\mathbf{L}}_t^c - offset$$

$$\tilde{\mathbf{L}}_t = \text{Concat}(\tilde{\mathbf{L}}_t^c, \tilde{\mathbf{L}}_t^l, \tilde{\mathbf{L}}_t^r) \tag{7}$$

$$\tilde{Class}_t = \text{MLP}_{cls}(\tilde{\mathbf{Q}}^{\mathbf{T}})$$

$$\tilde{T}_t = \text{MLP}_{bcls}(\tilde{\mathbf{Q}}^{\mathbf{T}})$$

$$\tilde{\mathbf{M}}_t = \text{Sigmoid}(\text{MLP}_{mask}(\tilde{\mathbf{Q}}^{\mathbf{T}}) \otimes \tilde{\mathbf{F}}_{bev}^t)$$

where $\mathbf{R}_{\mathbf{C}}^{\mathbf{S}}$ indicates the centerline reference points for stream queries. InSigmod refers to the inverse sigmoid function, while Denorm stands for denormalize. $offset$ represents the lateral distance from the centerline to both the left and right lanes. $\tilde{\mathbf{L}}_t$, $\tilde{Class}_t$, $\tilde{Type}_t$ and $\tilde{\mathbf{M}}_t$ are coordinates of lane segment, classes of centerline, boundary types and semantic BEV mask. Then, the transformation losses are represented as:

$$\mathcal{L}_{coord}^{Stream} = \mathcal{L}_{L1}(\tilde{\mathbf{L}}_t, \mathbf{L}_t)$$

$$\mathcal{L}_{cls}^{Stream} = \kappa_1 \mathcal{L}_{Focal}(\tilde{Class}_t, Class_t) + \kappa_2 \mathcal{L}_{CE}(\tilde{Type}_t, Type_t)$$

$$\mathcal{L}_{mask}^{Stream} = \kappa_3 \mathcal{L}_{CE}(\tilde{\mathbf{M}}_t, \mathbf{M}_t) + \kappa_4 \mathcal{L}_{Dice}(\tilde{\mathbf{M}}_t, \mathbf{M}_t) \tag{8}$$

$$\mathcal{L}_{query} = \kappa_5 \mathcal{L}_{coord}^{Stream} + \kappa_6 \mathcal{L}_{cls}^{Stream} + \kappa_7 \mathcal{L}_{mask}^{Stream}$$

where the values of $\kappa_1$, $\kappa_2$, $\kappa_3$, $\kappa_4$, $\kappa_5$, $\kappa_6$, and $\kappa_7$ are 1.0, 0.01, 1.0, 1.0, 0.025, 1.0 and 3.0.

### A.4 PREDICTION HEADS

After the decoder, we employ MLPs to predict lane segment coordinate, lane segment class, boundary class and BEV mask from the updated queries $\mathbf{Q}$:

$$\mathbf{R}_{\mathbf{c}} = \text{SigmoidMLP}_{pe}(\mathbf{PE})$$

$$\tilde{\mathbf{L}}^c = \text{MLP}_{reg}(\mathbf{Q}) + \text{InSigmod}(\mathbf{R}_{\mathbf{c}})$$

$$\tilde{\mathbf{L}}^c = \text{Denorm}(\text{Sigmoid}(\tilde{\mathbf{L}}^c))$$

$$offset = \text{MLP}_{offset}(\mathbf{Q})$$

$$\tilde{\mathbf{L}}^l = \tilde{L}^c + offset, \tilde{\mathbf{L}}^r = \tilde{\mathbf{L}}^c - offset$$

$$\tilde{\mathbf{L}} = \text{Concat}(\tilde{\mathbf{L}}^c, \tilde{\mathbf{L}}^l, \tilde{\mathbf{L}}^r) \tag{9}$$

$$\tilde{Class} = \text{MLP}_{cls}(\mathbf{Q})$$

$$\tilde{Type} = \text{MLP}_{bcls}(\mathbf{Q})$$

$$\tilde{\mathbf{M}} = \text{Sigmoid}(\text{MLP}_{mask}(\mathbf{Q}) \otimes \tilde{\mathbf{F}}_{bev})$$

$$\mathbf{Q}^{'} = \text{MLP}_{pre}(\mathbf{Q}), \mathbf{Q}^{''} = \text{MLP}_{suc}(\mathbf{Q})$$

$$\tilde{\mathbf{A}} = \text{Sigmoid}(\text{MLP}_{topo}(\text{Concat}(\mathbf{Q}^{'}, \mathbf{Q}^{''})))$$

where $\mathbf{R}_{\mathbf{c}}$ denotes centerline reference points and $\mathbf{PE}$ indicates positional embedding. $\tilde{\mathbf{A}}$ denotes the adjacency matrix that encodes the topological associations. The confidence threshold for the adjacency matrix is set at 0.5

### A.5 LANE SEGMENT LOSS

Lane segment loss is proposed by LaneSegNet (Li et al., 2023b):

$$\mathcal{L}_{ls} = \omega_1 \mathcal{L}_{vec} + \omega_2 \mathcal{L}_{seg} + \omega_3 \mathcal{L}_{cls} + \omega_4 \mathcal{L}_{type} + \omega_5 \mathcal{L}_{topo} \tag{10}$$

**Table 1:** Comparison of computational costs.

| Method | mAP | $AP_{ls}$ | $AP_{ped}$ | $TOP_{lsls}$ | FPS | FLOPs | Param. | Training memory cost | inference memory cost |
|---|---|---|---|---|---|---|---|---|---|
| LaneSegNet | 32.6 | 32.3 | 32.9 | 25.4 | 14.7 | 639.1G | 30.9M | 21.8GB | 4.0GB |
| FASTopoWM | 37.4 | 36.4 | 38.4 | 29.6 | 11.4 | 671.0G | 45.9M | 22.5GB | 4.2GB |

where the hyperparameters are defined as: $\omega_1 = 0.025$, $\omega_2 = 3.0$, $\omega_3 = 1.5$, $\omega_4 = 0.01$, and $\omega_5 = 5.0$. $\mathcal{L}_{vec}$ is the L1 loss computed between the predicted vectorized lanes and the ground truth lanes. $\mathcal{L}_{seg} = \mathcal{L}_{ce} + \mathcal{L}_{dice}$ consists of a Cross-Entropy loss and a Dice loss used to supervise the BEV semantic mask. The classification losses $\mathcal{L}_{cls}$ and $\mathcal{L}_{type}$ are used for lane segment classification. $\mathcal{L}_{topo}$ is a focal loss applied to supervise the topological relationship prediction.

### A.6 HUNGARIAN MATCHING

The cost function and weighting scheme used for the fast–slow system remain identical during parallel training. The cost function is defined as:

$$\mathcal{L}_{matching} = \beta_1 \mathcal{L}_{vec} + \beta_2 \mathcal{L}_{seg} + \beta_3 \mathcal{L}_{cls} + \beta_4 \mathcal{L}_{type} \tag{11}$$

where $\mathcal{L}_{vec}$, $\mathcal{L}_{seg}$, $\mathcal{L}_{cls}$, and $\mathcal{L}_{type}$ follow the same definitions as in Eq. 10. The weights $\beta_1$, $\beta_2$, $\beta_3$, and $\beta_4$ are set to 0.025, 3.0, 1.5, and 0.01, respectively.

### A.7 STREAMING TRAINING

We adopt the streaming training strategy for temporal fusion. For each training sequence, we randomly divide it into 2 splits at the start of each training epoch to foster more diverse data sequences. During inference, we use the entire sequences. Suppose a batch contains N samples, each from a different scene, read in chronological order. Temporal fusion is performed by determining whether the current data and the previously read data belong to the same scene. To facilitate temporal fusion, we introduce several memory modules, including stream query memory, stream BEV memory, and stream reference point memory, which store the predictions from the preceding frame.

### A.8 COMPUTATIONAL COST ANALYSIS

Tab. 1 shows the comparison of computational costs between our baseline method and our method. During inference on the OpenLane-V2 Subset A, our baseline model LaneSegNet requires 639.1G FLOPs and has 30.9M parameters. FASTopoWM integrates the powerful fast-slow branches and two world models on top of this baseline, resulting in 671.0G FLOPs and 45.9M parameters. As a trade-off, our method exhibits a slight decrease in FPS in exchange for a 4.6% mAP and a 4.2% $TOP_lsls$ improvement over the baseline. Additionally, the GPU memory usage of the baseline is 21.8 GB during training and 4.0 GB during inference per GPU card when the batch size is set to 2, while for FASTopoWM, it is 22.5GB and 4.2GB. The increase in training memory footprint for FASTopoWM is due to the parallel training of the dual-branch decoder. However, the inference memory usage of FASTopoWM is very close to that of the baseline.

### A.9 FRONT-VIEW VISUALIZATION

Fig. 1 presents the front-view visualization results. FASTopoWM detects the intersection faster than the temporal baseline model while maintaining temporally consistent and accurate detection results. In contrast, the baseline method exhibits poor temporal consistency in detection, leading to missed detections.

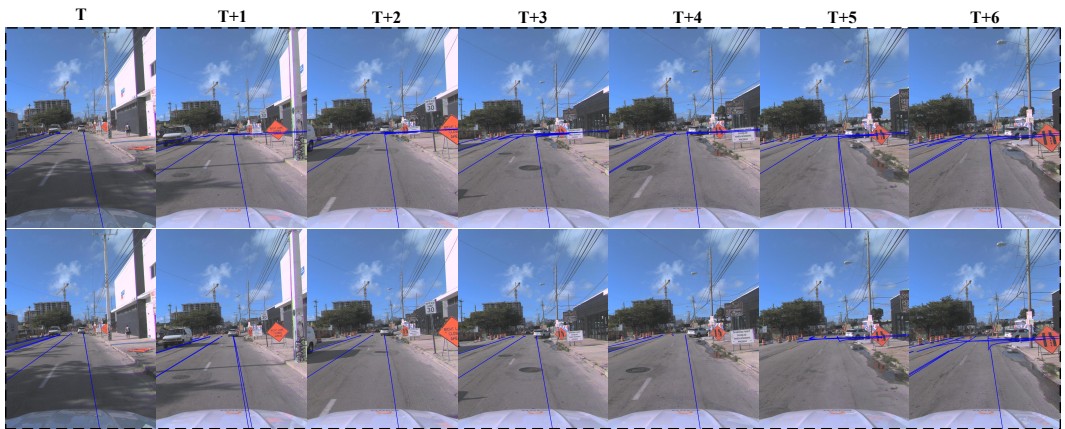

**Figure 1:** Front-view visualization of baseline and our FASTopoWM. The results of FASTopoWM are shown on the top, and the results of temporal baseline are shown on the bottom. For better viewing, zoom in on the images.

## A.10 DEMO

See the supplementary material vis.gif file for details. The visualization results demonstrate that our predictions maintain robust temporal consistency, reflected in the stable alignment of lane segment coordinates and topological structures as the ego vehicle moves.

## A.11 LIMITATION AND FUTURE WORK

World models demonstrate strong capability in predicting the next-timestep BEV and transforming query information for common driving scenarios, such as stationary states, straight-line driving, and gentle turns. However, their generalization may be inadequate for rare scenarios involving rapid changes in ego-vehicle pose, such as high-curvature turns. To address this, we plan to introduce noise to simulate diverse ego-vehicle pose variations, thereby enhancing the temporal transformation capability of world models in such corner cases. Furthermore, we intend to integrate Vision Language Models (VLMs) to aggregate world model outputs for improved detection, and leverage linguistic descriptions to enhance lane topology reasoning and interpretability. Finally, we aim to establish rules based on lane topology to provide actionable safety recommendations for autonomous driving.

## A.12 USE OF LLM

In this paper, Large Language Model is used only for writing enhancement purposes.

