# OpenReview forum: "FASTopoWM: Fast-Slow Lane Segment Topology Reasoning with Latent World Models"
_ICLR.cc/2026/Conference — Submitted to ICLR 2026_

### Official Review · Reviewer_S193 · 2025-10-24

**Soundness:** 2
**Presentation:** 2
**Contribution:** 3
**Rating:** 6
**Confidence:** 2

**Summary:**

This paper proposes FASTopoWM, a novel fast-slow lane segment topology reasoning framework augmented with latent world models, which decouples the network into dual pathways: a slow pipeline that leverages temporal information to enhance detection performance and a fast pipeline that performs single-frame perception for system robustness. The unified architecture enables parallel supervision of both historical and newly initialized queries, facilitating mutual reinforcement between the two systems, and introduces latent query and BEV world models conditioned on ego-motion to effectively capture temporal dynamics and enable robust state propagation. Extensive experiments on the OpenLane-V2 benchmark demonstrate that FASTopoWM achieves state-of-the-art performance in both lane segment detection and centerline perception.

**Strengths:**

1. **Novelty**: This paper is the first to introduce a fast–slow system into the field of lane-segment topology reasoning, leveraging latent world models to address the limitations of previous methods—specifically, their vulnerability to pose estimation failures and insufficient temporal propagation.

2. **quality**: The method and experiments are well-designed and complete. Comprehensive visualizations and released code make the paper more solid and reproducible.

3. **Experiment**: The proposed method achieves state-of-the-art performance on the OpenLane-V2 benchmark.

**Weaknesses:**

1. **About the novelty claim**: This paper appears to propose a method that improves upon existing stream-based approaches, as illustrated in Figure 1. However, the lane-segment topology reasoning task is not limited to stream-based methods. Therefore, Figure 1, the abstract, and the introduction should be revised to more clearly highlight the distinctions and advantages of this work compared with previous methods in the field.

2. **About the visualization**: The current visualizations only present BEV (bird’s-eye view) results. Showing the lane-segment predictions in the front-view images would better demonstrate the effectiveness of the proposed method. It is recommended to include visualizations from the surrounding-view images for a more comprehensive presentation.

**Questions:**

See weaknesses.

---

> ### Author Response · Authors · 2025-11-17
> **Response to Reviewer S193**
>
> ### Author Response
>
> We sincerely thank the reviewers for their recognition of our work. Based on the valuable feedback provided, we have made substantial revisions to the manuscript. We kindly invite the reviewers to refer to the updated version when reviewing our responses. Below, we provide point-by-point clarifications addressing the reviewers’ concerns.
>
> ---
>
> #### **1. Novelty Claim**
> Our method primarily involves significant modifications in the temporal modeling compared to existing approaches, so Figure 1 effectively illustrates the differences from temporal methods. Therefore, we have made corresponding revisions in the abstract and introduction. Please refer to **Lines 013–018** in the abstract and **Lines 046–053** in the introduction. Specifically, we summarize the main innovations of existing works, such as **topology graph modeling**[1], **endpoint alignment**[2], and **multi-attribute learning**[3]. However, these methods largely overlook **temporal modeling**, which is **crucial** for autonomous driving in dynamic environments. Therefore, our paper introduces a **world-model-based temporal framework**.
>
> Regarding temporal methods, we identify **three key limitations**:
> (1) **over-reliance on historical queries**,
> (2) **vulnerability to pose estimation failures**, and
> (3) **weak temporal propagation**.
>
> To address these issues, we propose a **unified framework** that integrates the **fast–slow system** with **world models**. Extensive and comprehensive experiments further demonstrate the effectiveness of our approach.
>
> ---
>
> #### **2. Front-view Visualization**
> In the revised manuscript, we have added front-view visualizations. Please refer to **appendix A.9 and Figure 1**, with the corresponding text highlighted in blue in **Lines 913–917**. Specifically, FASTopoWM detects intersections earlier than the temporal baseline model while maintaining temporally consistent and accurate detection results. In contrast, the temporal baseline method shows poor temporal consistency, resulting in missed detections.
>
> Thank you for the reviewers' comments, which have helped improve the presentation of our paper.
>
> ---
>
> [1] T2sg: Traffic Topology Scene Graph for Topology Reasoning in Autonomous Driving, CVPR 2025.
>
> [2] Topologic: An Interpretable Pipeline for Lane Topology Reasoning on Driving Scenes, NIPS 2024.
>
> [3] LaneSegNet: Map Learning with Lane Segment Perception for Autonomous Driving, ICLR 2024.

---

> > ### Comment · Reviewer_S193 · 2025-11-27
> >
> > I appreciate the author's efforts in addressing my concerns regarding novelty and visualization, and my doubts have also been resolved.
> >
> > Overall, I believe this is a meaningful work in the field of autonomous driving topology reasoning. The author also made great efforts in the rebuttal stage. Therefore, I maintain my positive evaluation and increase my confidence to 5.

---

> > > ### Author Response · Authors · 2025-11-27
> > >
> > > Thank you very much for your thoughtful feedback and for taking the time to reassess our work. We sincerely appreciate your positive evaluation and your increased confidence score. We are glad that our responses have effectively addressed your concerns regarding novelty, visualization, and the overall contribution of the work. Your recognition further motivates us to continue improving the clarity and impact of our research. Thank you again for your constructive comments and support.

---

### Official Review · Reviewer_xDds · 2025-10-29

**Soundness:** 3
**Presentation:** 2
**Contribution:** 3
**Rating:** 6
**Confidence:** 4

**Summary:**

This paper focuses on some issues arising from temporal bev-feature and query propagation in the lane prediction task. In order to alleviate the mentioned problems, the authors propose a framework that can leverage parallel supervision to historical BEV features and instance queries for mutual reinforcement, mainly by sharing the whole model and using the Latent World model to reconstruct the pseudo current features from past features. The extensive experiments demonstrate the effectiveness of the proposed components.

**Strengths:**

1. The parallel supervision for historical and current features via the latent world model to achieve better feature learning is novel and interesting.
2. The individual designs in the whole framework are reasonable to achieve the goal. And there are corresponding ablations to validate the effectiveness.

**Weaknesses:**

1.    Experiments on three claims in the Introduction are missing: (1) over-reliance on historical queries (2) vulnerability to pose estimation failures and (3) Weak temporal propagation. I would like to see some experiments, an analysis or even some cues on how the author identifies these problems.
2.    Likewise to problem 1, it’s still unclear to me why the “slow-fast system” can solve these problems fundamentally. Some evidence or statistics should be given.
3.    The definition of the Slow & Fast system in this paper may be confusing; originally, the Slow & Fast system on VLA is to address the asynchrony between fast model inference and slow decision processes in real-world action execution. While in this paper, ‘slow system’ and ‘fast system’ are more like a history feature aggregator and a current frame detector.
4.    Why, generally speaking, methods with Temporal feature aggregation perform worse than those without that, as shown on Table1, and 2?
5.    On Table 3, the author presents ablation studies on QWM and FWM, I would like to see the result that retains the QWM and FWM structure but removes the latent supervision (eq(3) and eq(4)).

**Questions:**

The questions have been listed on the points of weakness.

---

> ### Author Response · Authors · 2025-11-17
> **Response to Reviewer xDds (part I)**
>
> ### Author Response
>
> We sincerely thank the reviewers for their recognition of our work. Based on the valuable feedback provided, we have made substantial revisions to the manuscript. We kindly invite the reviewers to refer to the updated version when reviewing our responses. Below, we provide point-by-point clarifications addressing the reviewers’ concerns and correcting misunderstandings.
>
> ---
>
>
> #### **1. Experiments Demonstrating the Limitations of Stream-based Frameworks**
> In the revised manuscript, **Tables 3 and 4(h)** present the limitations of existing stream-based methods. We have added a detailed comparison in **Lines 462–471**, where the corresponding text is highlighted in red.
>
> **Table3**: Ablation studies on different modules. The baseline is the non-temporal framework LaneSegNet. *Stream* refers to the baseline augmented with temporal propagation.  *FS* denotes the fast-slow system. *QWM* and *BWM* represent the query world model and BEV world model, respectively. *Tem.* and *Sin.* indicate temporal and single-frame detection.
> | Modules | | | | Tem. | | Sin. | |
> |:---:|:---:|:---:|:---:|:---:|:---:|:---:|:---:|
> | *Stream* | *FS* | *QWM* | *BWM* | mAP | TOP$_{lsls}$ | mAP | TOP$_{lsls}$ |
> | | | | | - | - | 32.6 | 25.4 |
> | ✓ | | | | 34.0 | 28.1 | 29.8 | 23.4 |
> | ✓ | ✓ | | | 35.3 | 28.4 | 33.2 | 25.8 |
> | ✓ | ✓ | ✓ | | 36.3 | 29.0 | 33.7 | 26.3 |
> | ✓ | ✓ | | ✓ | 36.4 | 29.1 | 33.6 | 26.2 |
> | ✓ | ✓ | ✓ | ✓ | **37.4** | **29.6** | **34.1** | **26.6** |
>
> **Table4(h)**: Different temporal propagation modules.
> | Method | mAP | AP_{ls}| AP_{ped}| TOP_{lsls} |
> |:---:|:---:|:---:|:---:|:---:|
> | Baseline+Warp | 34.0 | 33.3 | 34.7 | 28.1 |
> | Baseline+WMs | **36.5** | **35.6** | **37.4** | **28.9** |
>
>
> Specifically, as shown in **Table 3 (rows 1 and 2)**, integrating the baseline with stream-based temporal propagation improves performance by 1.4% mAP and 2.7% TOP_{lsls}. However, its single-frame detection performance without pose supervision drops by 2.8% mAP and 2.0% TOP_{lsls}. This occurs because the temporal framework tends to over-rely on historical queries. When temporal propagation becomes unavailable due to missing pose information, the model falls back to single-frame detection, where its localization ability is weakened because it depends heavily on the initialized queries as anchors.  By integrating the fast–slow system with world models **(rows 3–6)**, we address the performance degradation issue of single-frame detection within temporal frameworks. This is because we decouple the supervision of initialized queries from that of historical queries, allowing the initialized queries to receive direct supervision from the ground truth. Our proposed model achieves substantial improvements, exceeding the baseline by 3.4% and 1.5% mAP in temporal and single-frame detection, respectively. Therefore, our method addresses **Limitation 1: over-reliance on historical queries** and **Limitation 2: vulnerability to pose estimation failures**.
>
> Furthermore, in **Table 4(h)**, we compare the stream-based warping approach with our world-model-based temporal propagation. The combination of baseline + world models outperforms baseline + warp by 2.5% mAP and 0.8 % TOP_{lsls}.  In addition, as shown in **Figure 4** in the main text and **Figure 1** in the appendix, our method exhibits much stronger temporal consistency, demonstrating that it also resolves **Limitation 3: weak temporal propagation.**
>
> ---
>
>
> #### **2. The Definition of Fast-Slow System**
> We added explanations in ***Lines 176--184** to clarify our definition of the **fast-slow system**. Specifically, many existing approaches implement fast and slow systems as **separate frameworks** and switch between them based on scenario conditions. In contrast, our method integrates both pathways into a **single model** without requiring additional networks. These pathways are trained under **parallel supervision**, enabling **mutual reinforcement** and **improving robustness**. Our architecture is closer to a **unified fast-slow thinking model** [1][2], where switching between pathways is controlled by a **trigger mechanism**, such as scenario difficulty. For example, using **slow thinking** with reasoning capabilities in challenging scenarios, and **fast thinking** in simpler ones. In our case, the trigger is the reliability of the **pose estimation** (details provided in the appendix).

---

> > ### Author Response · Authors · 2025-11-17
> > **Response to Reviewer xDds (part II)**
> >
> > In addition, the **fast–slow** system is closely related to our proposed **world models**. We have added the definition of **world models** in **Lines 198–211** of the revised manuscript. Specifically, we modify the world-model formulation by shifting the prediction sequence backward along the timeline, enabling the model to predict the **current state** based solely on **historical data**. The world models operate exclusively within the **slow pathway**, enhancing **temporal consistency**, while the fast pathway is used when pose information is unavailable, ensuring the **overall robustness** of the system. Both pathways benefit from each other during training. As seen in **rows 1 and 6 of Table 3**, our method surpasses the baseline by 1.5% in mAP and 1.2% in TOP_{lsls} for single-frame detection. As shown in **rows 2 and 3 of Table 3**, introducing the fast–slow system yields a 1.3% improvement in temporal mAP. This improvement originates from the **parallel supervision** applied to both pathways.
> >
> > ---
> >
> >
> >
> > #### **3. The Performance of temporal methods**
> > This is because methods such as **StreamMapNet**[3] and **SQD-MapNet**[4] are **not designed specifically** for lane segment detection; rather, they are developed for general HD map detection tasks. They do not incorporate dedicated designs that account for the **multiple attributes** of lane segments. When retraining these models for lane segment detection, we did not modify their architectures and only replaced the prediction heads to output lane segments and topology, in order to ensure a **fair comparison**. As a result, for the more complex topology structures involved in lane segment detection, their performance may be inferior to single-frame detectors that are explicitly tailored for lane segment modeling, such as **LaneSegNet**[5].
> >
> > ---
> >
> > #### **4. The Effect of Latent Supervision**
> > In **Table 4(g)** of the revised manuscript, we present the effect of **latent supervision**. Removing latent supervision leads to a clear performance drop.
> >
> >
> > **Table 4(g)**: Effect of latent supervision.
> >
> > | Method | mAP | AP_{ls}| AP_{ped} | TOP_{lsls} |
> > |:---:|:---:|:---:|:---:|:---:|
> > | FASTopoWM (without latent) | 36.6 | 35.8 | 37.5 | 29.0 |
> > | FASTopoWM (with latent) | **37.4** | **36.4** | **38.4** | **29.6** |
> >
> > ---
> >
> > [1] AutoVLA: A Vision-Language-Action Model for End-to-End Autonomous Driving with Adaptive Reasoning and Reinforcement Fine-Tuning, NIPS 2025.
> >
> > [2] Fast-slow Thinking for Large Vision-Language Model Reasoning, arXiv 2025.
> >
> > [3] StreamMapNet: Streaming Mapping Network for Vectorized Online HD map Construction, WACV 2024.
> >
> > [4] Stream Query Denoising for Vectorized HD-Map Construction, ECCV 2024
> >
> > [5] LaneSegNet: Map Learning with Lane Segment Perception for Autonomous Driving, ICLR 2024.

---

> ### Author Response · Authors · 2025-11-27
> **A Warm Reminder Regarding Our Previous Response**
>
> Dear Reviewer xDds,
>
> I hope this message finds you well.
>
> This is a gentle follow-up regarding your review of our submission. We sincerely appreciate the time and effort you have dedicated to the process.
>
> Please let us know if you require any additional information or clarifications from our side.
>
> Thank you for your consideration.
>
> Best regards,
>
> The Authors of Submission 351

---

> > ### Comment · Reviewer_xDds · 2025-11-27
> > **Response to the authors' rebuttal**
> >
> > Thanks for the detailed explanations from the authors. Most concerns are addressed. I tend to keep my score.

---

### Official Review · Reviewer_wPaF · 2025-10-31

**Soundness:** 2
**Presentation:** 3
**Contribution:** 1
**Rating:** 2
**Confidence:** 4

**Summary:**

FASTopoWM introduces a fast–slow framework with latent world models for temporal lane segment topology reasoning in BEV. The slow pipeline leverages temporal cues by transforming historical queries and BEV features into current “stream” representations via transformer-based query and BEV world models conditioned on relative pose (action latent), while the fast pipeline performs single-frame perception. A unified decoder with shared weights enables parallel supervision of stream and newly initialized queries, mitigating over-reliance on history and providing a robust fallback when pose estimates are unreliable. The BEV world model is trained self-supervised with MSE on adjacent-frame BEV features; the query world model uses transformation losses on coordinates, classes, and masks. On OpenLane‑V2, FASTopoWM reports state-of-the-art results: 37.4% mAP for lane segments and 46.3% OLS for centerlines, outperforming temporal and single-frame baselines.

**Strengths:**

- The paper is well written and clearly organized, and figures are effectively created to support the content.
- Experimental results verify the effectiveness of the proposed approach against several baselines.

**Weaknesses:**

**Major Weaknesses**

1. Substantial overlap with TopoStreamer[1] in problem framing and technical pipeline, without the necessary systematic comparison or discussion. TopoStreamer also targets temporal lane segment topology reasoning and essentially adopts the same or highly similar streaming mechanisms and supervisory objectives. Yet the manuscript provides no systematic analysis of differences or pros/cons. This is a serious issue by ICLR standards: when facing the most directly related, recent work on the same task, the authors neither include experimental comparisons nor articulate methodological distinctions and trade-offs. In fact, the two papers share highly consistent setups, so adding a direct comparison should be straightforward.
2. The main text lacks detailed quantitative and qualitative experiments to substantiate the claimed “three critical limitations in stream-based frameworks,” providing only final accuracy comparisons. This obscures the paper’s contribution and makes it difficult to assess the effectiveness of the proposed approach.
3. Ablations are insufficient: there is no systematic cross-study of K values, memory length, or training strategies (e.g., pure world model vs. conventional Warp+Fuse). Meanwhile, compared with other baseline frameworks, the proposed method exhibits a noticeable increase in computational overhead. More compute metrics, including FLOPs and memory usage, should be reported to provide a more comprehensive evaluation.

**Minor Weaknesses**

1. Figure 4 caption: “The results of TopStreamer are shown on the top, and the results of temporal baseline are shown on the bottom.” This appears to have mistakenly included another method’s name in this paper’s visualization caption.
2. Figure 1 caption: “Comparsion” is misspelled.
3. Several implementation details remain unclear (e.g., the exact cost functions and weights used for Hungarian matching when the fast and slow branches run in parallel, and whether sharing weights between the two branches during training exacerbates mode collapse).

[1] Yang Y, Luo Y, He B, et al. TopoStreamer: Temporal Lane Segment Topology Reasoning in Autonomous Driving[J]. arXiv preprint arXiv:2507.00709, 2025.

**Questions:**

Please see major weakness.

---

> ### Author Response · Authors · 2025-11-17
> **Response to Reviewer wPaF (part I)**
>
> ### Author Response
>
> We sincerely thank the reviewers for their recognition of our work. Based on the valuable feedback provided, we have made substantial revisions to the manuscript. We kindly invite the reviewers to refer to the updated version when reviewing our responses. Below, we provide point-by-point clarifications addressing the reviewers’ concerns and correcting misunderstandings.
>
> ---
>
> #### **1. Comparison with TopoStreamer**
> In the revised manuscript, we have added a detailed comparison with TopoStreamer. Please refer to Lines 134–145 (blue text) for details. In addition,we have added the performance comparisons with TopoStreamer in Tables 1 and 2.
>
> Specifically, TopoStreamer designs **temporal propagation**, **denoising objectives**, and **positional encoding injection** for multiple lane-segment attributes, enabling temporal perception of lane segments. Compared with TopoStreamer, our approach adopts a **stronger dual world-model design** for temporal propagation, using pose information as guidance to achieve more robust temporal perception. While TopoStreamer still relies on the traditional warping-based temporal propagation, it proposes **streaming attribute constraints** to ensure consistent propagation across multiple lane-segment attributes. It also introduces **attribute-aware denoising objectives** to improve the reliability of multi-pattern lane-segment associations. In addition, our paper identifies the **significant performance degradation** that occurs when temporal perception falls back to single-frame detection, and proposes a **fast–slow system** to enhance both temporal and single-frame detection performance. This design ensures system reliability in practical applications, and constitutes one of the **key innovations** that distinguishes our method from TopoStreamer.
>
> ---
>
> #### **2. Experiments Demonstrating the Limitations of Stream-based Frameworks**
> In the revised manuscript, Tables 3 and 4(h) present the limitations of existing stream-based methods. We have added a detailed comparison in Lines 462–471, where the corresponding text is highlighted in red.
>
> **Table3**: Ablation studies on different modules. The baseline is the non-temporal framework LaneSegNet. *Stream* refers to the baseline augmented with temporal propagation.  *FS* denotes the fast-slow system. *QWM* and *BWM* represent the query world model and BEV world model, respectively. *Tem.* and *Sin.* indicate temporal and single-frame detection.
> | Modules | | | | Tem. | | Sin. | |
> |:---:|:---:|:---:|:---:|:---:|:---:|:---:|:---:|
> | *Stream* | *FS* | *QWM* | *BWM* | mAP | TOP$_{lsls}$ | mAP | TOP$_{lsls}$ |
> | | | | | - | - | 32.6 | 25.4 |
> | ✓ | | | | 34.0 | 28.1 | 29.8 | 23.4 |
> | ✓ | ✓ | | | 35.3 | 28.4 | 33.2 | 25.8 |
> | ✓ | ✓ | ✓ | | 36.3 | 29.0 | 33.7 | 26.3 |
> | ✓ | ✓ | | ✓ | 36.4 | 29.1 | 33.6 | 26.2 |
> | ✓ | ✓ | ✓ | ✓ | **37.4** | **29.6** | **34.1** | **26.6** |
>
> **Table4(h)**: Different temporal propagation modules.
> | Method | mAP | AP_{ls}| AP_{ped}| TOP_{lsls} |
> |:---:|:---:|:---:|:---:|:---:|
> | Baseline+Warp | 34.0 | 33.3 | 34.7 | 28.1 |
> | Baseline+WMs | **36.5** | **35.6** | **37.4** | **28.9** |
>
>
> Specifically, as shown in **Table 3 (rows 1 and 2)**, integrating the baseline with stream-based temporal propagation improves performance by 1.4% mAP and 2.7% TOP_{lsls}. However, its single-frame detection performance without pose supervision **drops** by 2.8% mAP and 2.0% TOP_{lsls}. This occurs because the temporal framework tends to **over-rely on** historical queries. When temporal propagation becomes **unavailable** due to **missing pose information**, the model falls back to single-frame detection, where its localization ability is weakened because it only depends on the initialized queries as anchors.  By integrating the fast–slow system with world models **(rows 3–6)**, we **address the performance degradation issue** of single-frame detection within temporal frameworks. This is because we decouple the supervision of initialized queries from that of historical queries, allowing the initialized queries to receive direct supervision from the ground truth. Our proposed model achieves substantial improvements, exceeding the baseline by 3.4% and 1.5% mAP in temporal and single-frame detection, respectively. Therefore, our method addresses **Limitation 1: over-reliance on historical queries** and **Limitation 2: vulnerability to pose estimation failures**.
>
> Furthermore, in **Table 4(h)**, we compare the stream-based warping approach with our world-model-based temporal propagation. The combination of baseline + world models outperforms baseline + warp by 2.5% mAP and 0.8 % TOP_{lsls}. In addition, as shown in **Figure 4** in the main text and **Figure 1** in the appendix, our method exhibits much stronger temporal consistency, demonstrating that it also resolves **Limitation 3: weak temporal propagation.**

---

> ### Author Response · Authors · 2025-11-17
> **Response to Reviewer wPaF (part II)**
>
> ---
>
>
> #### **3. More Ablations**
> We thank the reviewer for the helpful suggestion. In the revised manuscript, we have added ablation studies on the **choice of
> K**, **memory length**, and **training strategies**. Please refer to **Tables 4(e)**, **4(f)**, and **4(h)** for details.
>
> Specifically, **Table 4(e)** examines the effect of different memory lengths in the BEV world model. We transform BEV features from longer-term historical frames to the current frame using the BEV world model, average them, and then fuse the result with the BEV feature of the current frame. The results indicate that increasing the memory length provides only marginal performance gains.
>
> As shown in **Table 4(f)**, we investigate the influence of the number of propagated queries in the query world model. Experiments demonstrate that the configuration used in FASTopoWM—propagating the **top 30% (i.e., 66)** queries—achieves the best performance.
>
> Finally, **Table 4(h)** shows that compared with traditional warping-based temporal propagation, our world models enable significantly more robust temporal propagation, improving mAP by 2.5%.
>
>
> ---
>
>
>
> #### **4. Details for Hungarian Matching**
> We have added details regarding the Hungarian matching in the A.6 of appendix. The corresponding text is highlighted in green in **Lines 878–886**.
>
> The cost function and weighting scheme used for the fast and slow system remain identical during parallel training. The cost function is defined as:
>
>  $L_{matching}$ = $\beta_1$ $L_{vec}$ + $\beta_2$ $L_{seg}$ + $\beta_3$ $L_{cls}$ + $\beta_4 $ $L_{type}$
> where  $L_{vec}$  is the L1 loss computed between the predicted vectorized lanes and the ground truth lanes. $L_{seg}$ = $L_{ce}$ + $L_{dice} $consists of a Cross-Entropy loss and a Dice loss used to supervise the BEV semantic mask. The classification losses $L_{cls}$ and $L_{type}$ are used for lane segment classification. The weights $\beta_1$, $\beta_2$, $\beta_3$, and $\beta_4$ are set to 0.025, 3.0, 1.5, and 0.01, respectively.
>
> **Table 1**: Comparison of computational costs.
> | Method | mAP | AP_{ls} | AP_{ped} | TOP_{lsls} | FPS | FLOPs | Param. | Training memory cost | Inference memory cost |
> |:---:|:---:|:---:|:---:|:---:|:---:|:---:|:---:|:---:|:---:|
> | LaneSegNet | 32.6 | 32.3 | 32.9 | 25.4 | 14.7 | 639.1G | 30.9M | 21.8GB | 4.0GB |
> | FASTopoWM | 37.4 | 36.4 | 38.4 | 29.6 | 11.4 | 671.0G | 45.9M | 22.5GB | 4.2GB |
>
>
> ---
>
>
>
> #### **5. Comparison of Computational Costs**
> We have added details regarding the comparison of computational costs in the **A.8 of appendix**. The corresponding results are summarized in **Table 1** of the appendix, with the text highlighted in red in **Lines 899–910**.
>
> Specifically, our baseline model LaneSegNet requires **639.1G FLOPs** and has **30.9M parameters**. FASTopoWM integrates the **powerful fast-slow branches** and **two world models** on top of this baseline, resulting in **671.0G FLOPs** and **45.9M parameters**. As a trade-off, our method exhibits a slight decrease in **FPS** in exchange for a 4.6\% mAP and a 4.2\% TOP$_{lsls}$ improvement over the baseline. Additionally, the GPU memory usage of the baseline is **21.8 GB** during training and **4.0 GB** during inference per GPU card when the **batch size** is set to 2, while for FASTopoWM, it is **22.5GB** and **4.2GB**. The increase in training memory footprint for FASTopoWM is due to the **parallel training** of the dual-branch decoder. However, the inference memory usage of FASTopoWM is very close to that of the baseline.
>
>
> ---
>
>
>
> #### **7. Mode Collapse**
> In our weight-sharing implementation, the inputs to the fast and slow systems are concatenated and processed in parallel, and parallel supervision is applied accordingly. In our comprehensive experiments, we **did not observe any mode collapse**. On the contrary, the fast and slow systems consistently provide **mutual performance benefits**. As seen in **rows 1 and 6 of Table 3**, our method surpasses the baseline by 1.5% in mAP and 1.2% in TOP_{lsls} for single-frame detection. As shown in **rows 2 and 3 of Table 3**, introducing the fast–slow system yields a 1.3% improvement in temporal mAP. This improvement originates from the **parallel supervision** applied to both pathways.
>
>
> ---
>
>
> #### **8. Correction of Typographical Errors**
> We thank the reviewer for the careful examination of our manuscript. All of the typographical errors identified by the reviewer have been corrected.

---

> ### Author Response · Authors · 2025-11-27
> **A Warm Reminder Regarding Our Previous Response**
>
> Dear Reviewer wPaF,
>
> I hope this message finds you well.
>
> This is a gentle follow-up regarding your review of our submission. We sincerely appreciate the time and effort you have dedicated to the process.
>
> Please let us know if you require any additional information or clarifications from our side.
>
> Thank you for your consideration.
>
> Best regards,
>
> The Authors of Submission 351

---

### Official Review · Reviewer_Xfqc · 2025-11-04

**Soundness:** 2
**Presentation:** 2
**Contribution:** 2
**Rating:** 2
**Confidence:** 4

**Summary:**

In this work, the authors introduce a fast-slow dual system named FASTopoWM to solve the current models’ inability to effectively leverage temporal information. Although some stream-based temporal propagation methods are demonstrated to be effective, they still depend too much on historical queries, are vulnerable to pose estimation failures, and have insufficient temporal propagation. FASTopoWM solves these limitations by introducing the dual-system framework. It can simultaneously supervise both historical and new queries and prompt mutual benefits. Besides, they also introduce latent representations to further improve the slow system’s performance. FASTopoWM is verified on OpenLane-V2, and achieves state-of-the-art performance on both lane-segment detection and centerline detection.

**Strengths:**

- This work provides a much better way to incorporate historical information to facilitate lane topology reasoning, compared to StreamMapNet.
- FASTopoWM achieves SOTA performance on OpenLane-V2 with a 37.4 mAP (+3.8 compared to the previous SOTA baseline Topo2Seq) while maintaining a certainly acceptable latency (11.4)
- Ablations shown in Table 3 demonstrated various parts’ functions in FASTopoWM. The experiments are overall comprehensive.

**Weaknesses:**

- The paper's primary weakness is a disconnect between its terminology and the methods described. Certain concepts, like "world models" and the "fast-slow system," feel overstated.
- For example, the "world model" is a two-layer transformer that predicts features for the next timestep. This is a much simpler implementation than what is typically understood by the term.
- Similarly, the proposed "fast-slow system" primarily relies on the slow system's output. The fast system only acts as a fallback when "reliable pose information is missing or inaccurate" (Line 196), which diminishes its role compared to conventional dual-system architectures.

**Questions:**

This leads to a key question: What is the precise trigger for the fast system?
- How is "missing or inaccurate" pose information defined or measured?
- Is this fallback mechanism only used during initialization, or can it be triggered at any point during operation?

---

> ### Author Response · Authors · 2025-11-17
> **Response to Reviewer Xfqc (part I)**
>
> ### Author Response
>
> We sincerely thank the reviewers for their recognition of our work. Based on the valuable feedback provided, we have made substantial revisions to the manuscript. We kindly invite the reviewers to refer to the updated version when reviewing our responses. Below, we provide point-by-point clarifications addressing the reviewers’ concerns and correcting misunderstandings.
>
> ---
>
> #### **1. Definition of the world model**
>
> In our work, we employ **latent world models**[1], which are fundamentally different from **image-based world models** used for generating synthetic observations. As a result, our decoder can remain compact and simple. To further clarify our definition of latent world models, we have added additional explanations in **Lines 198--211** of the revised manuscript.
>
> Specifically, compared to traditional driving world models, our approach has **two key differences**.
> First, we shift the prediction sequence backward along the timeline, allowing the model to infer the **current** state from **historical** observations.
> Second, we adopt two **latent world models** that does not reconstruct raw observations from latent features. This enables the model to focus solely on temporal propagation in BEV space and in the query space, resulting in a more **compact architecture** and **reduced learning complexity** than **image-based world models** [2][3].
>
> Thus, our latent world model follows the workflow:
>
>
> $$ O^{T-1} -> F^{T-1}_{latent} -> F^{T}_{latent} ->  Lane Segments$$
>
>
> ---
>
> #### **2. Definition of the fast-slow system**
> We added explanations in **Lines 176--184** to clarify our definition of the **fast-slow system**. Specifically, many existing approaches implement fast and slow systems as **separate frameworks** and switch between them based on scenario conditions. In contrast, our method integrates both pathways into a **single model** without requiring additional networks. These pathways are trained under **parallel supervision**, enabling **mutual reinforcement** and **improving robustness**. Our architecture is closer to a unified **fast-slow thinking model** [4][5], where switching between pathways is controlled by a **trigger mechanism**, such as scenario difficulty. For example, using **slow thinking** with reasoning capabilities in challenging scenarios, and **fast thinking** in simpler ones. In our case, the trigger is the reliability of the **pose estimation** (details provided in the appendix).
>
> Importantly, the fast system is not merely a fallback used when the slow temporal system suffers degradation. During training, **effective supervision** on the fast-pathway queries significantly improves the performance of the slow (temporal) branch. As shown in **rows 2 and 3 of Table 3**, introducing the fast–slow system yields a 1.3% improvement in temporal mAP. This is because historical queries used in the temporal branch typically receive higher Hungarian matching scores than the initialization queries of single-frame detection, causing ground-truth assignments to favor historical queries unless the fast branch is properly trained. By incorporating world models to enhance temporal propagation, the slow system becomes more powerful, which in turn improves the single-frame detection capability of the fast system. As seen in **rows 1 and 6**, our method surpasses the baseline by 1.5% in mAP and 1.2% in TOP_$lsls$ for single-frame detection.
>
> **Table3**: Ablation studies on different modules. The baseline is the non-temporal framework LaneSegNet. *Stream* refers to the baseline augmented with temporal propagation.  *FS* denotes the fast-slow system. *QWM* and *BWM* represent the query world model and BEV world model, respectively. *Tem.* and *Sin.* indicate temporal and single-frame detection.
> | Modules | | | | Tem. | | Sin. | |
> |:---:|:---:|:---:|:---:|:---:|:---:|:---:|:---:|
> | *Stream* | *FS* | *QWM* | *BWM* | mAP | TOP$_{lsls}$ | mAP | TOP$_{lsls}$ |
> | | | | | - | - | 32.6 | 25.4 |
> | ✓ | | | | 34.0 | 28.1 | 29.8 | 23.4 |
> | ✓ | ✓ | | | 35.3 | 28.4 | 33.2 | 25.8 |
> | ✓ | ✓ | ✓ | | 36.3 | 29.0 | 33.7 | 26.3 |
> | ✓ | ✓ | | ✓ | 36.4 | 29.1 | 33.6 | 26.2 |
> | ✓ | ✓ | ✓ | ✓ | **37.4** | **29.6** | **34.1** | **26.6** |
>
>
> ---

---

> ### Author Response · Authors · 2025-11-17
> **Response to Reviewer Xfqc (part II)**
>
> #### **3. Pose estimation and Trigger**
>
>
> In the appendix (Lines 706--771), we describe in detail why pose estimation may be inaccurate or unavailable in practice and how this affects the trigger that switches between the fast and slow pathways.
>
> Specifically, poses provide dynamic information for autonomous vehicles and are typically obtained via **GPS (Global Positioning System)** and **IMU (Inertial Measurement Unit)**. Specifically, GPS determines position through satellite-based ranging (**absolute coordinates**), while the IMU derives position and orientation by integrating acceleration and angular velocity (**relative motion**). The advantage of GPS is its absence of long-term cumulative errors, whereas the IMU offers high frequency. The integration of these two methods for pose estimation has been widely adopted in many public datasets, like **KITTI** [6], **ApolloScape** [7], **Cityscapes** [8], **H3D** [9], etc. However, this approach also has **limitations**. For instance, in the **KITTI** dataset, the localization system is susceptible to GPS signal interruptions, particularly in **urban canyons** or **tunnels** where interference is common [10]. In such cases, the IMU can be used for compensation, but it suffers from **error accumulation** that increases exponentially over time (drift), resulting in poor stability. In the **nuScenes** [11], to mitigate this issue, carefully collected detailed **high-definition LiDAR point cloud maps** are employed. However, in practice, it is **not feasible** to create such point cloud maps for all scenarios.
>
> ### *How to define missing or inaccurate pose?*
> From a localization and sensor-fusion perspective, it refers to cases where the estimated vehicle position or orientation (pose) cannot be reliably determined due to **signal loss**, **sensor drift**, or **inconsistency between different measurements**. It can be characterized and measured differently for **GPS** and **IMU** systems.
>
> #### **GPS.** Pose information is considered missing or inaccurate when the satellite-based positioning solution fails to meet required quality thresholds. In our practical experience, commonly observed issues include: **1. RTK (Real-Time Kinematic) status deteriorates from FIX → FLOAT → DGPS → Single.** This indicates a gradual degradation in positioning accuracy. **2. Poor geometric precision.** High HDOP (Horizontal Dilution of Precision)/PDOP (Position Dilution of Precision) values (e.g., HDOP $>$ 4) reflect poor satellite geometry. **3. Weak or unstable satellite signals.** Low C/N$_{o}$ (carrier-to-noise ratio $<$ 30 dB-Hz) or fewer than 6 visible satellites.}
>
> #### **IMU.** Pose is considered inaccurate when integration of accelerations and angular rates accumulates error beyond acceptable limits. Since IMU data are relative and drift over time, accuracy degrades when not corrected by GPS or other sensors. In our practical experience, commonly observed issues include: **1. Rapid drift of position or orientation.** A drift is identified if the position drift meets or exceeds 0.1 m per second without external correction, or if the yaw/roll bias accumulation meets or exceeds 0.1° per minute. **2. Dynamic inconsistency.** Pose-derived velocity or acceleration inconsistent with wheel odometry or CAN bus data. **3. Time synchronization error.**
>
> ### *How to measure inaccurate pose?*
>
>
> To evaluate the accuracy of the current pose estimated from **IMU** and **GPS**, one can perform consistency and residual checks across the three data sources—**CAN bus** (vehicle kinematics), **IMU**, and **GPS**. The core idea is to compare physical quantities derived from the estimated pose with independent measurements from the vehicle and sensors. **1. Consistency between **IMU** and **CAN bus**.** If the **IMU** is well calibrated, its measured yaw-rate and acceleration should match the CAN yaw-rate and acceleration within small tolerance. **2. GPS residual analysis.** When fusing **GPS** and **IMU**, we can validate pose accuracy by examining the residual between the fused result and the raw GPS measurement:
>
>
> $$ r = P_{GPS} - P_{fusion} $$
> where $P_{GPS}$ is the observed GPS position and P$_{fusion}$is the predicted position obtained from **IMU** integration combined with the previous state estimate. We then determine whether the magnitude of r exceeds a predefined threshold.

---

> ### Author Response · Authors · 2025-11-17
> **Response to Reviewer Xfqc (part III)**
>
> #### **Trigger.** **GPS** plays a primary role in positioning because it also provides the reference needed to correct the **IMU**. Consider the following scenario: before entering a **tunnel**, **GPS** and **IMU** fusion operates normally. Inside the **tunnel**, **GPS** becomes **unavailable** and the system relies solely on the **IMU**, causing the **drift to grow over time**. After exiting the **tunnel**, **GPS** becomes available again and corrects the accumulated **IMU drift**, restoring normal behavior. Therefore, we monitor both **GPS signal quality** and **GPS residuals** as **triggers** for detecting inaccurate pose.
>
> In traditional temporal methods, when pose information becomes unreliable, the system must **fall back** to **single-frame detection** and **stop temporal propagation**. In this case, as validated in **Table3 (rows 1 and 2)**, the performance of the single-frame detector drops by 2.8\% in mAP and 2.0\% in TOP$_{lsls}$ compared with a well-trained single-frame model.
>
>
> Alternatively, an autonomous driving system would need to carry both temporal and single-frame models, which introduces **additional computational and memory overhead**. However, with our fast--slow architecture that shares weights and performs parallel supervision for both single-frame and temporal branches, this problem is resolved. Moreover, the single-frame detection performance is improved, achieving gains of approximately 1.5\% in mAP and 1.2\% in TOP$_{lsls}$ **(rows 1 and 6)**.
>
> In addition, introducing the world model further boosts the temporal detection performance, improving the baseline temporal method by 3.4\% in mAP and 1.5\% in TOP$_{lsls}$ **(rows 2 and 6)**.
>
> ---
>
> #### **4. Fallback mechanism.**
> At **initialization**, no historical frames are available, so the first frame is processed using the **fast pathway**. For subsequent frames, temporal information becomes available, and inference switches to the **slow pathway**.
>
> When pose estimation becomes unreliable—such as in **dense urban areas**, **tunnels**, or **remote regions**—the trigger switches inference to the **fast system**. Once pose quality is restored, the system switches back to the **slow temporal pathway**.
>
> This reflects the typical **real-world workflow**:
> before entering a tunnel → slow system; inside the tunnel (GPS loss) → fast system; after exiting → slow system again.
>
> ---
> [1] End-to-End Driving with Online Trajectory Evaluation via BEV World Model, ICCV 2025.
>
>
> [2] Drivedreamer: Towards Real-World-Drive World Models for Autonomous Driving, ECCV 2024.
>
> [3] Gaia-1: A Generative World Model for Autonomous Driving, arXiv 2023.
>
> [4] AutoVLA: A Vision-Language-Action Model for End-to-End Autonomous Driving with Adaptive Reasoning and Reinforcement Fine-Tuning, NIPS 2025.
>
> [5] Fast-slow Thinking for Large Vision-Language Model Reasoning, arXiv 2025.
>
> [6] Are We Ready for Autonomous Driving? The KITTI Vision Benchmark Suite, CVPR 2012.
>
> [7] The ApolloScape Dataset for Autonomous Driving, CVPR Workshop 2018.
>
> [8] The Cityscapes Dataset for Semantic Urban Scene Understanding, CVPR 2016.
>
> [9] The H3D Dataset for Full-surround 3D Multi-Object Detection and Tracking in Crowded Urban Scenes, ICRA 2019.
>
> [10] AI-IMU dead-reckoning, IEEE TIV 2020.
>
> [11] Nuscenes: A Multimodal Dataset for Autonomous Driving, CVPR 2020.

---

> > ### Comment · Reviewer_Xfqc · 2025-11-27
> >
> > Thanks for your rebuttals, and many of my concerns are addressed. I will accordingly raise my score. For the fallback mechanism, do you have any statistical numbers indicating how frequently your fast systems normally step in?

---

> ### Author Response · Authors · 2025-11-27
> **A Warm Reminder Regarding Our Previous Response**
>
> Dear Reviewer Xfqc,
>
> I hope this message finds you well.
>
> This is a gentle follow-up regarding your review of our submission. We sincerely appreciate the time and effort you have dedicated to the process.
>
> Please let us know if you require any additional information or clarifications from our side.
>
> Thank you for your consideration.
>
> Best regards,
>
> The Authors of Submission 351

---

> ### Author Response · Authors · 2025-11-28
>
> Thank you very much to the reviewer for the valuable comments and support.
>
> We conducted an internal statistical analysis based on a subset of our privately collected dataset.
> We selected approximately **500 driving clips**, each containing **30 frames**, covering a diverse set of driving environments, including:
>
> - **Urban open roads** (broad city streets with moderate traffic)
> - **Urban canyon areas** (dense high-rise buildings)
> - **Rural areas** (remote countryside and forest regions)
> - **Highways and multi-layer overpasses**
> - **Tunnels**
>
>
> The statistical results are summarized as follows:
>
> ---
>
> | **Scene Type**                  | **Probability of Drift** | **Typical Causes** |
> |--------------------------------|---------------------------|--------------------|
> | Urban Open Roads               | **5.3%**                  | Mild occlusion, light multipath |
> | Urban Canyon (High-Rise Areas) | **18.7%**                 | Heavy multipath, limited sky visibility |
> | Rural Areas                    | **22.4%**                 | Tree canopy, terrain blockage |
> | Highways / Overpasses          | **17.9%**                 | Bridge structures, layered interference |
> | Tunnels                        | **92.1%**                 | Complete GNSS loss |
>
>
> ---
>
> ### **Case Analysis Across Different Environments**
>
> #### **1. Urban Open Roads**
> **Typical reasons:**
> - Temporary occlusion
> - Light multipath near mid-rise buildings
>
> #### **2. Urban Canyons**
> **Typical reasons:**
> - Significant multipath from glass facades
> - Narrow sky visibility causing unstable satellite geometry
>
> #### **3. Rural Areas**
> **Typical reasons:**
> - Dense tree canopy attenuating GNSS reception
> - Terrain variations blocking satellite line-of-sight
>
> #### **4. Highways / Multi-layer Overpasses**
> **Typical reasons:**
> - Overpass decks temporarily blocking GNSS
> - Ambiguity between stacked highway layers
>
> #### **5. Tunnels**
> **Typical reasons:**
> - Full GNSS signal loss
> - Long tunnel sections leading to IMU drift accumulation
>
> ---
>
> In these situations, accumulated pose estimation errors can degrade the reliability of the slow system.
> When such drift is detected, the system **automatically switches to the fast system** to maintain robust real-time performance.
>
> We sincerely appreciate the reviewer’s thoughtful comments, which help highlight the importance and practicality of our dual-system design under challenging localization conditions.

---

### Author Response · Authors · 2025-11-24
**Global response to Area Chair**

We sincerely thank the Area Chair for overseeing the review process and all reviewers for their thoughtful and constructive feedback. In response to the comments from the four reviewers, we have undertaken **substantial revisions** to the manuscript. Below, we summarize the major improvements made throughout the paper:

---

### **1. Clearer Definitions and Conceptual Clarifications**

- We refined the definitions of the **latent world model** (Lines 198–211) and the **fast–slow system** (Lines 176–184), directly addressing conceptual misunderstandings raised by multiple reviewers.

---

### **2. Strengthened Motivation and Improved Novelty Statement**

- We revised both the **abstract** (Lines 013–018) and **introduction** (Lines 046–053) to better articulate our contributions and highlight the critical limitations of prior temporal methods—including heavy dependence on historical queries, susceptibility to pose failures, and weak temporal propagation.
- We emphasized how our unified world-model-driven temporal framework fundamentally resolves these limitations.

---

### **3. Expanded Experiments and Additional Comparative Studies**

In response to reviewers’ requests, we significantly broadened our experimental evaluation:

- Added **comparisons with TopoStreamer** in text (Lines 134–145) and in Tables 1 & 2.
- Included **new ablations** on memory length, number of propagated queries (K), and training strategies (Tables 4(e), 4(f), 4(h)).
- Added **new comparison** between world-model–based propagation and warping-based propagation (Table 4(h)).
- Added **analysis revealing limitations of stream-based methods** (Tables 3 and 4; Lines 462–471).
- Added **latent supervision ablation** (Table 4(g)).
- Added detailed explanation of **Hungarian matching** (Appendix A.6).
- Expanded **computational cost analysis** including memory, FLOPs, and FPS comparisons (Appendix A.8; Table 1).
- Added **front-view visualizations** (Appendix A.9 and Figure 1).
- Additionally, we evaluated our private dataset and provided the **empirical switching frequency** of the fast–slow system.

---

### **4. Addressing Practical Concerns: Pose Estimation & Robustness**

- We provided a detailed explanation of **pose definition**, its reliability, and how **triggering mechanisms** operate (Appendix Lines 706–771).
- We added practical criteria for pose unreliability detection (GPS/IMU residuals, signal quality metrics) and explained the interaction between the fast and slow pathways under real-world conditions.
- We demonstrated how our design mitigates degradation when temporal methods fall back to single-frame detection.

---

### **5. Improved Structure, Clarity, and Error Corrections**

- We polished the writing, reorganized several sections for clarity, and corrected all typographical errors identified by the reviewers.
- We updated and refined figures and tables for improved readability and consistency.

---

### **6. Reviewers’ Responses to Our Rebuttal**

After reviewing our rebuttal:



- **Reviewer Xfqc** stated that *“many of my concerns are addressed”*, **increased the score**, and further asked about the fast–slow system switching frequency.  This indicates a consistently **positive attitude** toward improving the score. We provided a prompt response to the new query, although the reviewer was unable to follow up further because they were prohibited from posting additional comments.
- **Reviewer xDds** confirmed that *“most concerns are addressed”* and maintained a **positive** evaluation.
- **Reviewer S193** explicitly acknowledged that their concerns regarding novelty, visualization, and technical clarity were fully resolved, and further stated that *“this is a meaningful work in the field of autonomous driving topology reasoning”*. The reviewer maintained a **positive recommendation** and **raised their confidence to 5**.
- **Reviewer wPaF** initially had misunderstandings about our method. We addressed their concerns point-by-point with extensive additional experiments and explanations. While this reviewer did not respond further because they were prohibited from posting additional comments. We believe the clarifications and the consistent positive evaluations from the other reviewers will also shape their final assessment positively.

---

### **Summary**

Overall, we have implemented **major conceptual clarifications, significantly expanded experiments, strengthened comparisons**, and improved both the clarity and organization of the manuscript. These revisions comprehensively address the reviewers’ concerns and substantially enhance the technical depth, practical relevance, and overall presentation of our work.

Following the rebuttal, reviewers expressed **positive views** of the paper, with reviewers increasing their **scores** and **confidence levels**.

We respectfully invite the Area Chair to consider these comprehensive improvements in the final assessment of our submission.

---

### Meta-Review · Area_Chair_m42Y · 2026-01-04

**Summary:**

This paper received mixed scores (6, 6, 2, 2). The reviewers’ concerns mainly focus on the novelty and definitions of the world model and the fast–slow (double) system (Xfqc, xDds, S193), insufficient discussion of related work (wPaF), and limitations in ablation studies and visualizations (wPaF, xDds, S193).

The authors provided a comprehensive rebuttal and made a clear effort to address the reviewers’ concerns. Most of the issues were addressed through clarification, additional experiments, and improved presentation. However, the definition and evaluation of the “world model” and the fast–slow system remain aspects that warrant further exploration and validation.

**Reviewer Concerns:**

1. Overstated concepts of “world model” and “fast–slow system.” Although the authors refined the definition of the latent world model in the rebuttal, it remains debatable whether a module that provides BEV features for lane segments can be reasonably referred to as a “world model.” In addition, the definition of the fast–slow system may need to be reconsidered, as it primarily functions as a fallback mechanism and a training-time assistant rather than a fully symmetric dual-system design (Xfqc).

2. Fast–slow system robustness and generalization. The authors provided additional ablation studies to demonstrate the effectiveness of the fast–slow system, including results in Table 3 of the rebuttal and analyses of the triggering mechanism of the dual system. However, the robustness of this system is not yet fully demonstrated. In particular, the trigger based on inaccurate pose estimation deserves further investigation. It remains unclear whether the chosen threshold is generalizable across different scenarios. Moreover, the robustness of the system could be better supported through qualitative or failure-case analyses (e.g., challenging scenarios such as tunnels), rather than relying solely on aggregate quantitative metrics.

**Reviewer Scores:**

Xfqc: 2 → 4 or 5. The reviewer explicitly raised their score during the discussion period.

wPaF: 2 → 2. Although TopoStreamer was included in the revised comparisons, the reviewer may still considered the ablation studies to be insufficient.

xDds: 6 → 6. The reviewer maintained their positive score.

S193: 6 → 6. The reviewer appreciated the contribution of the fast–slow system and the added visualizations.

---

### Decision · Program_Chairs · 2026-01-26

Reject